# Modelled surface climate response to Icelandic effusive volcanic eruptions: Sensitivity to season and size

Tómas Zoëga[1], Trude Storelvmo[1], and Kirstin Krüger[1]

[1]Department of Geosciences, University of Oslo, Oslo, Norway

**Correspondence:** Tómas Zoëga (tomas.zoega@geo.uio.no) and Kirstin Krüger (kirstin.kruger@geo.uio.no)

**Abstract.** Effusive, long-lasting volcanic eruptions impact climate through emission of gases and subsequent production of aerosols. Previous studies, both modelling and observational, have made efforts in quantifying these impacts and untangling them from natural variability. However, due to the scarcity of large and well observed effusive volcanic eruptions, our understanding remains patchy. Here we use an Earth system model to systematically investigate the climate response to high-latitude, effusive volcanic eruptions, similar to the 2014-15 Holuhraun eruption in Iceland, as a function of eruption season and eruptive size. The results show that the climate response is regional and strongly modulated by different seasons, with mid-latitude cooling during summer and Arctic warming during winter. Furthermore, as eruptions become larger in terms of sulfur dioxide emissions, the climate response becomes increasingly insensitive to variations in the emission strength, levelling out for eruptions between 20 and 30 times the size of the 2014-15 Holuhraun eruption. Volcanic eruptions are generally considered to lead to surface cooling, but our results indicate that this is an oversimplification, especially in the Arctic where we find warming to be the dominating response during fall and winter.

## 1 Introduction

Volcanic eruptions vary greatly in their behaviour. Some are dominated by explosive activity where the magma explodes and is erupted as tephra. In other cases, explosive activity is mostly absent and the magma is mainly erupted as lava. Eruptions falling into the latter group are referred to as effusive eruptions. Their emissions stay close to the ground, mostly in the lower and middle troposphere. They release various gaseous species, with water vapour, carbon dioxide, and sulfur dioxide being the most prominent (e.g., Textor et al., 2004). Of those, sulfur dioxide ($SO_2$) is the most relevant for short term climate impacts as it is a precursor to sulfate ($SO_4$) aerosols (Robock, 2000). These aerosols mainly impact climate through interactions with radiation, either directly (Graf et al., 1998) or indirectly as cloud condensation nuclei (CCN) through various aerosol-cloud interactions (Gassó, 2008).

Previous studies have observed a shortwave radiative forcing due to aerosol-cloud interactions (more numerous cloud droplets and higher cloud albedo with increased CCN concentrations) (Twomey, 1977) as a result of effusive volcanic emissions. Examples include the 2008 and 2018 Kilauea eruptions in Hawaii (Eguchi et al., 2011; Breen et al., 2021), the 2012 Mount Curry eruption in the South Sandwich Islands (Schmidt et al., 2012), and the 2014-15 Holuhraun eruption in Iceland (Gettelman et al., 2015; McCoy and Hartmann, 2015; Malavelle et al., 2017). Adjustments to aerosol-cloud interactions (Al-

brecht, 1989) have also been identified, as there is evidence for a significant increase in cloud cover during the first months of the 2014-15 Holuhraun eruption (Chen et al., 2022), as well as during the 2008 and 2018 Kilauea eruptions (Chen et al., 2024). In our previous study (Zoëga et al., 2023), we demonstrate with observational data, reanalysis, and model simulations that the 2014-15 Holuhraun eruption led to surface warming in the Arctic in the early winter of 2014-15 through increased cloud cover and increased liquid water path and subsequent trapping of longwave radiation under limited sunlight.

Iceland is volcanically active with an average of 20 to 25 eruptions per century during the historical period, covering the past ∼1100 years. These eruptions have varied vastly in size and characteristics, with roughly one out of every five being either effusive or mixed effusive-explosive (Thordarson and Larsen, 2007). Examples include the 1783-84 Laki eruption, which is estimated to have emitted a total of 122 Tg $SO_2$ over a period of eight months (Thordarson and Self, 2003), and the 939-940 Eldgjá eruption which emitted around 220 Tg $SO_2$ over a period of at least 1.5 years (Thordarson et al., 2001; Oppenheimer et al., 2018; Hutchison et al., 2024). The Great Thjorsa lava eruption (around 8000 years before present) is thought to have been the largest effusive eruption on Earth during the Holocene, with a lava production of at least 21 km$^3$ (Hjartarson, 1988; Siebert et al., 2010). As a reference, the lava production of the 1783-84 Laki and 939-940 Eldgjá eruptions amounted to about 15 km$^3$ and 20 km$^3$ respectively (Thordarson and Self, 1993; Thordarson et al., 2001; Sigurðardóttir et al., 2015). Closer in time is the aforementioned 2014-15 Holuhraun eruption which emitted up to 9.6 Tg $SO_2$ over a period of six months (Pfeffer et al., 2018) and produced about 1.2 km$^3$ of lava (Bonny et al., 2018). Icelandic volcanoes do, therefore, have a history of very large, long-lasting effusive eruptions.

It is only for the past few decades that we have been able to accurately monitor high-latitude volcanic eruptions and their climate impacts, namely since the beginning of the satellite era (Robock, 2000; Carn et al., 2016). The focus has mostly been on explosive eruptions (Haywood et al., 2010; Kravitz et al., 2010; Andersson et al., 2015) and their climate impacts have been revealed to highly depend on factors such as the eruption latitude, season and size, the emission altitude, and the atmospheric background state (e.g., Schneider et al., 2009; Kravitz and Robock, 2011; Toohey et al., 2019; Zambri et al., 2019; Marshall et al., 2020; Fuglestvedt et al., 2024; Zhuo et al., 2024). Despite considerable research efforts in recent years, the climate impacts of high-latitude effusive eruptions remain less understood, particularly how they relate to environmental and eruptive parameters. Here we address this issue using an Earth system model and systematically investigate the climate response to idealized high-latitude, long-lasting effusive volcanic eruptions as a function of eruption season and emission strength.

## 2  Methods

### 2.1  Model

We simulate the climate response to a range of effusive volcanic eruptions using the Community Earth System Model version 2.1.3 with the Community Atmosphere Model version 6, referred to as CESM2(CAM6) (Danabasoglu et al., 2020). It has 32 vertical levels which extend to an altitude of 2.26 hPa (ca. 40 km). For horizontal resolution we use 0.9° latitude by 1.25° longitude. All of our simulations are coupled with active atmosphere, ocean, sea ice, and land components.

CESM2(CAM6) includes a simplified sulfur chemistry scheme, described by Barth et al. (2000), which simulates both gas-phase and aqueous oxidation of $SO_2$ into $SO_4$. The atmospheric oxidants ozone ($O_3$) and the hydroxyl radical (OH), along with stratospheric aerosols, are prescribed from CESM2 historical CMIP6 simulations using the Whole Atmosphere Community Climate Model (WACCM) (Gettelman et al., 2019). The Modal Aerosol Module (MAM4) (Liu et al., 2016) simulates the formation and development of tropospheric aerosols. The four log-normal aerosol modes of MAM4 are Aitken, accumulation, coarse, and primary carbon. Together they include sulfate, sea salt, primary and secondary particulate organic matter, black carbon, and soil dust, which are internally mixed within within each mode. The conversion of aerosol from one mode to another is simulated through coagulation and condensation (Liu et al., 2012, 2016). The second version of the Morrison-Gettelman scheme (MG2) (Gettelman and Morrison, 2015) is used for prognostic cloud microphysics.

CESM2(CAM6) includes the unified cloudy turbulent scheme Cloud Layers Unified By Binormals (CLUBB) (Golaz et al., 2002). In CLUBB, cloud entrainment processes which could lead to deceased LWP are controlled by prognostic vertical turbulent fluxes and a tunable air parcel entrainment rate. However, both LWP and cloud cover are relatively insensitive to variation in the CLUBB parameter representing the entrainment rate (Guo et al., 2015). In their modelling study (not using CLUBB), Karset et al. (2020) further found that other factors, such as the sensitivity of the autoconversion rate to cloud droplet number concentration, play an even larger role in controlling the LWP than parameterized entrainment processes.

## 2.2 Simulations

We carry out a transient control run, corresponding to the model years 2005-2015, using the CMIP6 historical forcing (Eyring et al., 2016). For the year 2015, extensions of the existing historical CMIP6 forcing fields were used when available (van Marle et al., 2017; Hoesly et al., 2018), otherwise the SSP2-4.5 forcing (O'Neill et al., 2016) was applied.

From the control run, we branch off a number of simulations where volcanic emissions are added. These branches are six months long and we refer to them as eruption simulations. For each scenario considered in this study (see below), this leads to ten eruption simulations, each of which has its own unique initial conditions.

The volcanic eruptions in our simulations are represented by prescribed $SO_2$ emissions. We construct a standard eruption scenario, using petrological estimates of emissions from the 2014-15 Holuhraun eruption as a reference (Thordarson and Hartley, 2015; Zoëga et al., 2023) (see Fig. 1). Emissions are highest during the first month and gradually decay afterwards. Daily emissions are constant within each month (as approximated by 30 days). We then modify this standard scenario to represent eruptions of different sizes. All our eruptions are located at the site of the 2014-15 Holuhraun eruption at 64.9°N and 16.8°W, they last for 180 days, and emissions are well mixed between 1 and 3 km above sea level.

We are interested in the climate impacts from eruptions of different sizes and therefore vary the strength of the volcanic emissions by multiplying the standard emission scenario in Fig. 1 with a range of scaling factors. In addition to the ×1 scaling factor, corresponding to a Holuhraun-sized eruption, we perform simulations using scaling factors of ×5 and ×25, covering the plausible range of Icelandic effusive eruptions, and ×50, extending into the size range of the largest known flood basalts on Earth (Kasbohm and Schoene, 2018). We are also interested in how different eruption seasons modulate the climate response and perform eruption simulations branched off from the control run at the first days of March, June, September, and December

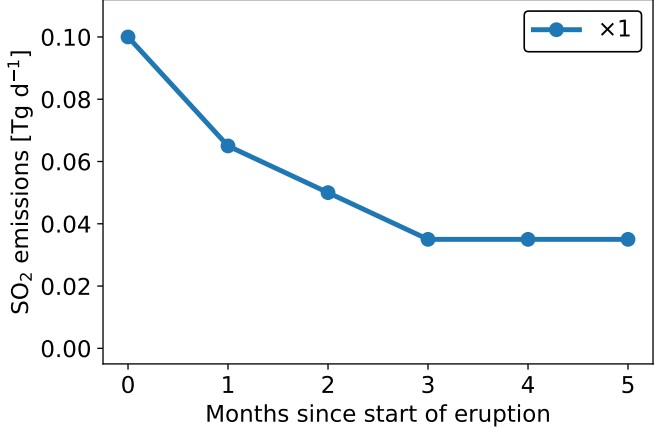

**Figure 1.** Volcanic $SO_2$ emissions rates for the standard eruption scenario ($\times 1$) used in this study. Daily emissions are constant within each month, well mixed between 1 and 3 km above sea level, and located at the cite of the 2014-15 Holuhraun eruption at 64.9°N and 16.8°W.

in each model year between 2005 and 2014. This results in ten eruption simulations for each combination of starting date and magnitude scaling. Throughout this study, we refer to those combinations with a scaling factor and a start month. For example, a $\times 5$ eruption starting in June is `x5jun`.

## 2.3 Observations and reanalysis

To supplement the model simulations, we look at data from the ERA5 reanalysis (Hersbach et al., 2020, 2024) and timeseries of observed surface air temperature. The observational timeseries are from Svalbard lufthavn/airport and Jan Mayen (NCCS, 2023), Danmarkshavn and Ittoqqortoormiit (DMI, 2023), and Grímsey (Icelandic Met Office, 2024).

## 2.4 Anomalies and significance

For a variable $y$ from our simulations, we calculate absolute anomalies such that

$$(\Delta y)_{\mathrm{abs}} = y_{\mathrm{erupt}} - y_{\mathrm{contr}} \tag{1}$$

and relative anomalies such that

$$(\Delta y)_{\mathrm{rel}} = \frac{y_{\mathrm{erupt}} - y_{\mathrm{contr}}}{y_{\mathrm{contr}}}. \tag{2}$$

This results in an ensemble of ten sets of anomalies for each combination of scaling factor and start month. The two simulations being compared (control and eruption) match on all background conditions (such as initial meteorology, background emissions,

greenhouse gas concentrations, etc.), and only differ on a single aspect, namely the volcanic $SO_2$ emissions. This approach is termed a matched-pairs analysis (e.g., Barlow, 1993).

For a measure of confidence, we calculate 95% confidence intervals (CI's) based on a two-tailed $t$-test such that

$$CI = \mu \pm t^* \cdot \hat{\sigma} \tag{3}$$

with $\mu$ being the ten member ensemble mean, $t^*$ an appropriate value from the $t$-statistics, and $\hat{\sigma} = \sigma/\sqrt{n}$ the standard error of the ensemble. Here $\sigma$ is the standard deviation of the ensemble and $n = 10$ the number of ensemble members.

For the observational timeseries and ERA5 reanalysis, anomalies are calculated with respect to a linear fit for the 30 year period 1984 to 2013. The constructed 95% prediction interval is calculated as $\pm 1.96\sigma$ from the mean of the detrended 1984 to 2013 timeseries, with $\sigma$ being the standard deviation of the timeseries.

## 2.5 Logarithmic fit and growth rate

To investigate the climate response as a function of eruption size, we fit a logarithmic curve to the anomalies $\Delta y$ such that

$$\Delta y_{\text{fit}} = a \ln(bx + 1) \tag{4}$$

where $x$ represents magnitude scaling factors, and $a$ and $b$ fitting coefficients. We calculate $a$ and $b$ using the method of least squares. A 1 is added to $bx$ to satisfy $\Delta y_{\text{fit}}(x = 0) = 0$. That is, no anomalies in the case of no eruption. We further calculate a growth rate (GR), which represents the relative change in $\Delta y_{\text{fit}}$ per magnitude scaling factor, such that

$$\text{GR} = \frac{1}{y} \cdot \frac{d}{dx}(\Delta y_{\text{fit}}) = \frac{1}{y} \cdot \frac{a}{x + 1/b}. \tag{5}$$

## 3 Results

Due to the high number of simulations performed in this study, we will focus on the `x5jun` and `x5dec` eruptive scenarios for illustrative purposes unless otherwise stated. We choose eruptions starting in June and December as we expect the climate response from summer and winter eruptions to generally represent the extremities on either end of the response spectrum. We choose the $\times 5$ scaling scenario as such eruptions are both very large and realistic, being approximately half way between the 2014-15 Holuhraun eruption and the 1783-84 Laki eruption in terms of mean $SO_2$ emission rate.

### 3.1 $SO_4$ aerosols and CCN

The conversion of $SO_2$ gas to $SO_4$ aerosols is controlled by the oxidation capacity of the atmosphere, which in turn depends to a large extent on sunlight availability. $SO_4$ aerosol production from precursor gases is therefore highly seasonal. This can clearly

be seen in our simulations where the volcanic aerosol load is much higher during the first three months of eruptions starting in June (June to August (summer), Fig. 2a) compared to the first three months of eruptions starting in December (December to February (winter), Fig. 2d). This seasonal difference is largest in the Arctic, as defined by the Arctic circle, where the aerosol load is $5.8 \pm 1.5$ times higher during summer than in winter in our $\times 5$ simulations.

$SO_4$ aerosols are very hygroscopic and therefore effective as CCN (e.g., Hobbs, 2000). In our simulations, the modelled $SO_4$ aerosol perturbations dominate the distribution of CCN (Figs. 2b and 2e), as evident when the spatial patterns of the aerosol and CCN anomalies are compared. In both cases, the dominant transport is toward north-east, namely over the Greenland and Norwegian Seas, northern Eurasia, and into the Arctic. We also see smaller, but significant, aerosol anomalies covering much larger areas, extending from the central North Atlantic, across North Africa and the Mediterranean Sea, across central Asia, and

all the way into the North Pacific and the Bering Sea. This applies for both summer and winter. The main difference between the seasons is the magnitude of the anomalies. The relative anomalies reveal a different pattern, especially in the case of the CCN (Figs. 2c and 2f). The greatest relative CCN anomalies occur in the Arctic, with up to 5-fold increase in summer and more than doubling in winter. The reason is the low background CCN level in the Arctic (Figs. A1a and A1d) (Choudhury and Tesche, 2023), which is a result of its relatively weak local CCN sources and the long distance from strong ones at lower

latitudes (e.g., Bigg and Leck, 2001).

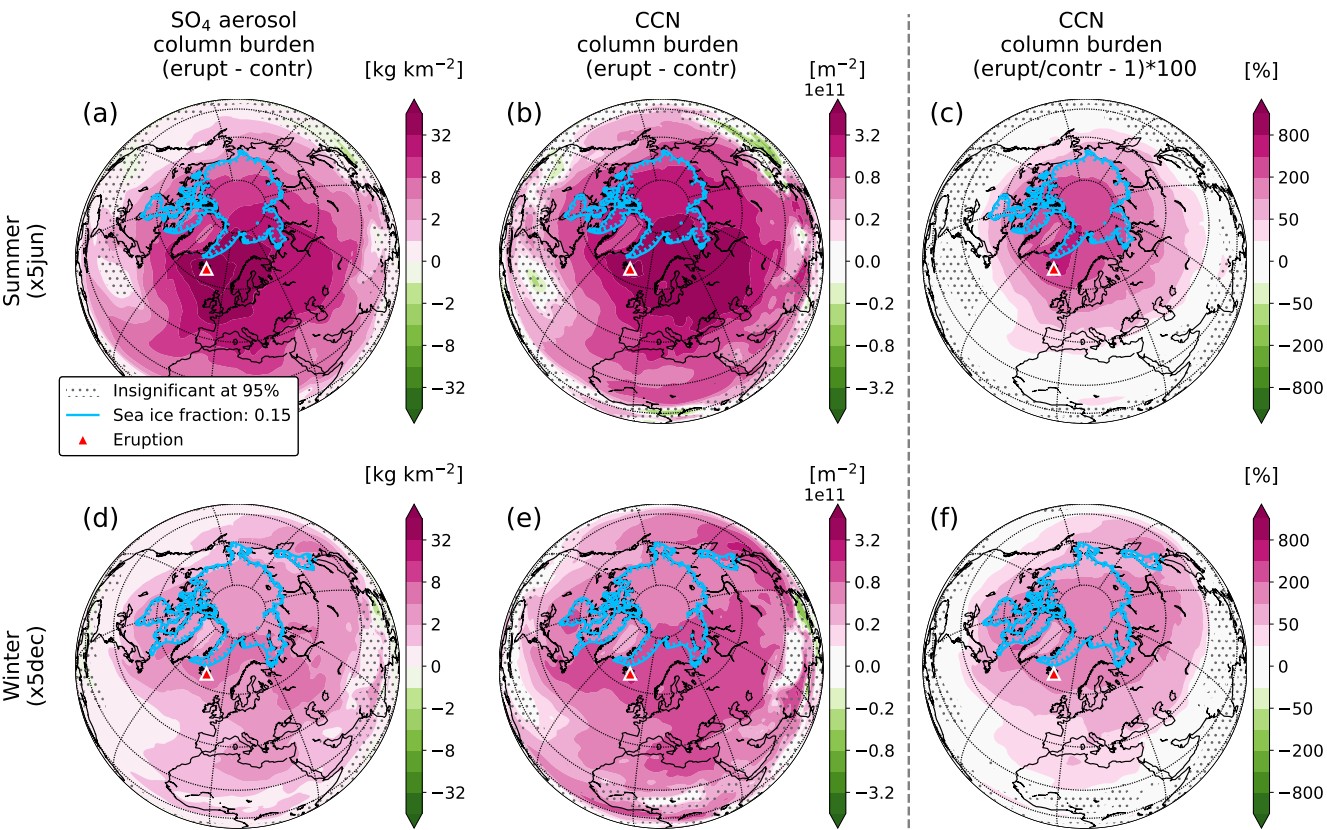

**Figure 2.** Ensemble mean absolute anomalies from the CESM2(CAM6) simulations for the first three months of the eruption for: The SO$_4$ aerosol column burden for the (a) `x5jun` and (d) `x5dec` scenarios, and the cloud condensation nuclei (CCN, at 0.1 % supersaturation) column burden for the (b) `x5jun` and (e) `x5dec` scenarios. To the right of the vertical dashed line are relative CCN column burden anomalies for (c) summer and (f) winter. The dotted regions indicate insignificance at the 95 % confidence level calculated with a two-tailed $t$-test, and the blue contours the mean sea ice edge for the first three months of the eruption from the eruption runs (15 % sea ice cover defines the sea ice edge). Summer refers to the June to August mean, and winter to the December to February mean.

## 3.2 Cloud droplets

Since SO$_4$ aerosols are effective as CCN, they can considerably alter cloud properties. Generally speaking, we expect a positive CCN perturbation to increase the number of cloud droplets and decrease their size (Twomey, 1977).

In the CCN poor Arctic, clouds are particularly sensitive to CCN perturbations. During the relatively warm and moist summer, this results in few but large cloud droplets (Figs. A1b and A1c respectively). In our eruption simulations, we see an increase in the number of cloud droplets (Fig. 3a), closely resembling the pattern of relative CCN increase. As expected, the cloud droplets also shrink considerably (Fig. 3b), especially over the Arctic sea ice where they are the largest in the control run. Contrary to the summer response, which is mainly in the Arctic, the largest cloud droplet anomalies during winter are found

over the Labrador Sea and the Sea of Okhotsk, followed by the open ocean areas off the Arctic sea ice edge in the Atlantic
sector. Cold air outbreaks from Canada, Siberia, and the Arctic sea ice transport cold and CCN poor air over open ocean,
leading to the formation of clouds with few but large droplets (Figs. A1e and A1f). These clouds are particularly sensitive to
CCN perturbations and respond strongly by increasing the number of droplets and decreasing their size (Figs. 3c and 3d).

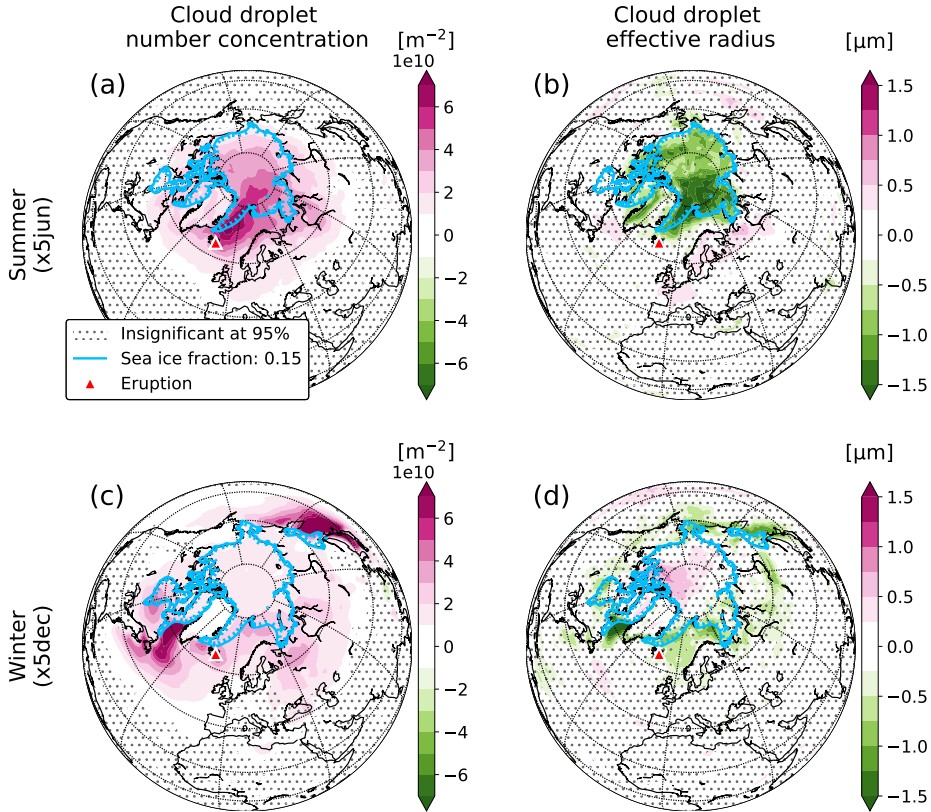

**Figure 3.** Same as Fig. 2 but for vertically integrated cloud droplet number concentration (a and c), and vertically averaged cloud droplet
effective radius (b and d)

### 3.3   Cloud lifetime

By increasing the number of cloud droplets and decreasing their size, CCN perturbations have the potential to affect the liquid
water content of clouds as well as their horizontal and vertical extent (Albrecht, 1989). We simulate a significant increase in
the cloud liquid water path (LWP), both in summer and winter (Figs. 4a and 4c respectively), which correlates well with the
increased cloud droplet number concentration.

In summer, this LWP increase is mainly bound to the Arctic. It can be explained by delayed precipitation through smaller
cloud droplets and slower collision-coalescence process over the sea ice in the central Arctic and suppressed precipitation

over the ice free Nordic Seas (Fig. A3b). Cloud cover over the Arctic remains unaffected (Fig. 4b) since the Arctic is mostly overcast in summer already (Fig. A2b) (Curry et al., 1996). However, we do simulate increased low level cloud cover over northern Europe where background cloud cover is lower than in the central Arctic.

Delayed or suppressed precipitation also explains the increased LWP in winter over the Labrador Sea and the Sea of Okhotsk (Fig. 4c) where we see a small but significant precipitation reduction (Fig. A3d), accompanied by a cooling signal (Fig. 5f). 170 This is, however, not the case in the central Arctic where the model shows a significant increase in the LWP despite the near absence of precipitating clouds (Fig. A3c). Instead we suggest the following.

Increased droplet number concentration at the edge of the Arctic basin leads to a local increase in LWP. This results in increased trapping of longwave radiation and subsequent surface warming (see Sections 3.4 and 3.5). This warming induces a deeper subpolar low in the North Atlantic with accompanying stronger southerly winds which advect warm air into the 175 Arctic (Fig. A11d). With a warmer Arctic, the liquid water content of ice-containing clouds increases, which results in larger liquid cloud droplets in the central Arctic in our simulations (Fig. 3d). It is well-established that when the ratio of liquid to ice water content in clouds increases, they generally precipitate less efficiently (the Wegener-Bergeron-Findeisen process) and their lifetimes increase (e.g., Tsushima et al., 2006; Storelvmo et al., 2011; Tan and Storelvmo, 2019), hence the increased cloud cover and LWP in the wintertime Arctic basin. This process is represented in CESM2(CAM6). The resulting surface 180 warming weakens the strong temperature inversion in the central Arctic (Fig. A5f), leading to increased updraft (Fig. A5e) and yet more cloud formation.

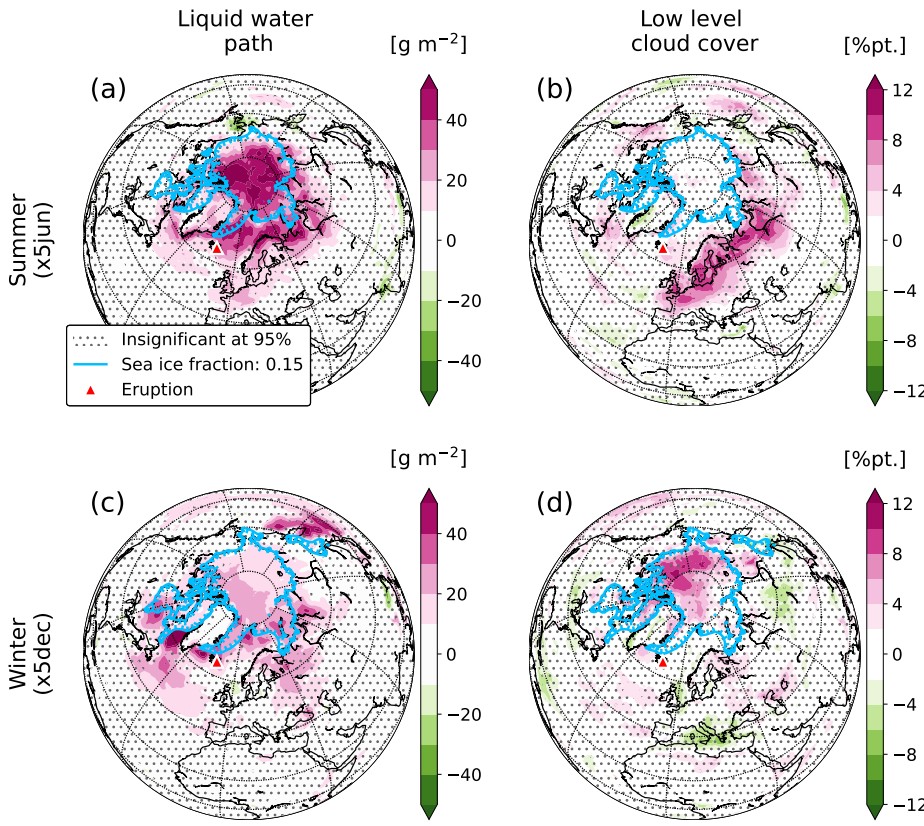

**Figure 4.** Same as Fig. 2 but for vertically integrated liquid water path (a and c) and low level cloud cover (b and d).

## 3.4 Surface radiation

For LWP $\gtrsim 30$ g m$^{-2}$, clouds become opaque in the longwave (LW) part of the radiative spectrum (Slingo et al., 1982; Shupe and Intrieri, 2004), meaning that once this threshold is passed, an increase in the liquid water content of clouds will not affect
their abilities to absorb and emit LW radiation. This is the case in our simulations in the Arctic during summer where the background LWP is about $140$ g m$^{-2}$ (Fig. A2a). As a result, the LW trapping abilities of the low level Arctic clouds, as represented by the downward LW flux at the surface (FLDS, Fig. 5a), only marginally increase in the summer months despite the considerable LWP increase. The winter is a different story. Then the mean LWP over the Arctic sea ice is about $40$ g m$^{-2}$, dropping below $25$ g m$^{-2}$ north of Greenland and Canada. The relatively modest LWP increase over the Arctic sea ice, along
with the increased low level cloud cover, therefore lead to a strong increase in the LW trapping of the clouds in that area. In our x5dec simulations, the model shows an Arctic mean December to February FLDS increase of almost $+8$ W m$^{-2}$, reaching to more than $+16$ W m$^{-2}$ in the central Arctic (Fig. 5d).

For the shortwave (SW) part of the radiative spectrum, radiative extinction increases with increased LWP for a much wider LWP range (e.g., Han et al., 1998; Glenn et al., 2020). Whereas absorption and re-emission dominate in the LW part of the

spectrum, scattering plays a major role for SW radiation, with smaller particles scattering more efficiently than larger ones (e.g., Fouquart et al., 1990). As a result, the model shows a strong decrease in downward SW flux at the surface (FSDS) across the entire Arctic and northern Europe during summer (Fig. 5b), closely coinciding with the increased LWP and decreased cloud droplet size. In our x5jun simulations, the model shows an Arctic mean June to August FSDS decrease of almost $-17$ W m$^{-2}$. During winter, sunlight is limited at high latitudes, and largely absent in the Arctic, and hence the model hardly

shows any SW anomalies (Fig. 5e).

Direct interactions between $SO_4$ aerosols and radiation are highly wavelength dependent. Whereas $SO_4$ aerosols barely affect LW radiative transfer, they effectively attenuate SW radiation, mainly through scattering (Kiehl and Briegleb, 1993; Clapp et al., 1997). In the Arctic, we would therefore expect direct aerosol effects to be the most effective during summer and negligible during winter. This is the case in our simulations. The model shows an increase in the summertime aerosol optical

depth at $550$ nm of around $0.5$ over the Greenland and Norwegian Seas (Fig. A6a), with anomaly patterns closely following the modelled volcanic $SO_4$ aerosol load. As a result, the clear sky component of the downward SW surface flux (FSDSC, Fig. A6b) plays a considerable role in SW radiative transfer during summer. Surface radiative fluxes therefore depend on both direct and indirect aerosol effects during summer whereas the indirect effects, that is aerosol-cloud-radiation interactions, dominate during winter.

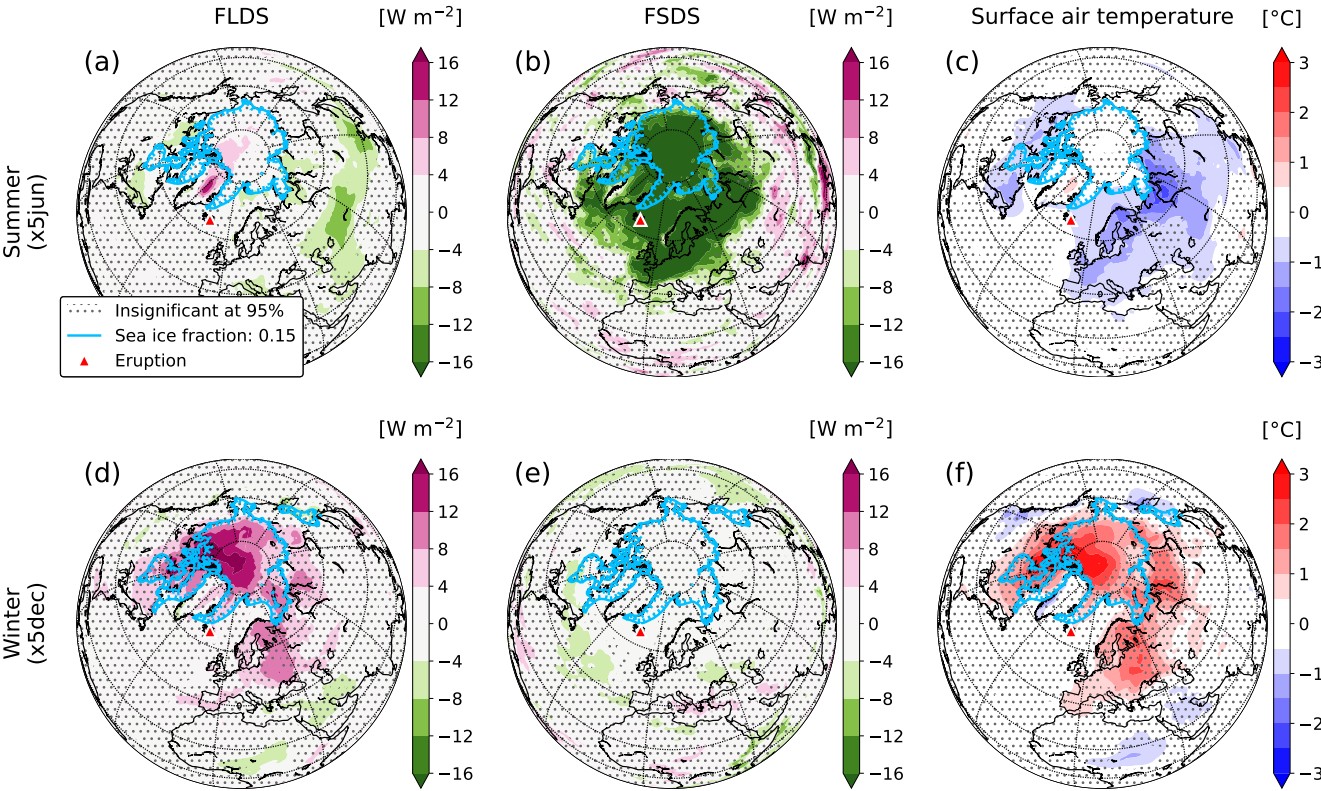

**Figure 5.** Same as Fig. 2 but for downward longwave radiative flux at the surface (FLDS) (a and d), downward shortwave radiative flux at the surface (FSDS) (b and e), and surface air temperature (c and f).

### 3.5 Surface air temperature

When it comes to surface air temperatures, the model shows pronounced warming in the Arctic during winter (Fig. 5f). This warming is widely significant and reaches more than $+3°C$ in the remote areas north of Canada and Greenland. The reason for this warming is the trapping of LW radiation under limited sunlight as a result of increased low level cloud cover and LWP as discussed in the previous sections.

During summer, there are significant cooling anomalies over northern Eurasia and North America, reaching more than $-2°C$ over Siberia. The model also shows a cooling of more than $-1°C$ over the Greenland, Norwegian, and Barents Seas. Interestingly, there is hardly any significant temperature response over the Arctic sea ice during summer. We interpret this to mainly be a result of the relatively high albedo of the sea ice and reduced sea ice melting.

Multiple reflections between low level clouds and the ground play an important role in SW surface radiative forcing. Where clouds cover bright surfaces, these reflections considerably reduce the effectiveness of the SW cloud shielding. This effect is well known and has been observed in the Arctic (Wendler et al., 1981). In our simulations, it is clearest during spring and early

summer where the model shows a small reduction in the net downward SW flux over the Arctic sea ice compared to the open ocean areas off the sea ice edge (Fig. A7b). The multiple reflection effect is especially sensitive to variations in ground albedo at high albedo values, with its effectiveness sharply decreasing during late summer as the Arctic sea ice fraction decreases.

Over dark surfaces, for example open ocean, these reflections play a minor role. Additionally, increased cloud shielding from direct sunlight decreases sea ice melt, leading to less heat being absorbed by the sea ice from the atmosphere (Fig. A8b).

Additionally, the model shows an increase in the Arctic sea ice fraction following the start of the `x5jun` eruptions (Fig. A8b). This increase is Arctic wide but most prominent outside of the central Arctic where the background sea ice fraction is between 50 % and 60 %. There the model shows an increase of up to +15 %pt. This indicates that the shielding effects of the

230 clouds slow down the sea ice melt during summer, making the Arctic surface more reflective and amplifying the SW reflection effect discussed above. The Arctic sea ice response during winter is not as widespread. However, the model shows a December to February decrease in sea ice fraction following the start of the `x5dec` eruptions of down to −10 %pt. along the sea ice edge in the Greenland, Barents, and Bering Seas (Fig. A8d).

### 3.6 Seasonal cycle

Until now we have focused on eruptions starting in summer and winter. To get a fuller picture of the seasonal cycle, we add simulations for eruptions starting in March (`x5mar`) and September (`x5sep`) and look at monthly means for the Arctic north of the Arctic circle (Fig. 6).

For the $SO_4$ aerosol load, the cloud droplet number concentration, the cloud droplet effective radius, and the LWP, the model shows clear seasonal variations with largest responses in summer and smallest in winter. The main reason is the pronounced

seasonality of $SO_4$ aerosol formation, which depends largely on available sunlight. The low level cloud cover displays the opposite behaviour, with anomalies being largest in winter and smallest in autumn. This has to do with the background conditions, as the Arctic is almost completely overcast during the summer months and hence only a small increase in cloud cover is to be expected.

In some instances, anomalies from different eruption scenarios are significantly different from each other despite covering the

245 same months. This is clearest for the aerosol anomalies. The reason for this is the gradual decay of emissions in our eruption scenarios, which results in less sulfur being available for aerosol formation as the eruption progresses. This has cascading effects which eventually lead to the apparent discrepancies in the cloud anomalies.

During mid-winter, there is a surface warming in the Arctic of up to +3°C. The confidence intervals are broad, indicating a large uncertainty in the magnitude of this warming. Despite this, the model shows significant warming in December and

250 January. In mid-summer, there is moderate cooling of down to −1°C. The summer cooling is more consistent among the different ensemble members compared to the winter warming, resulting in narrower confidence intervals. During fall (September-November), there is a discrepancy between the temperature responses of the `x5jun` and `x5sep` simulations, with cooling in the former and warming in the latter. Unlike the aerosol and cloud parameters discussed earlier, the main reason here is not the gradual decay of the volcanic emissions but a delayed response. During the first three months of the `x5jun` eruptions,

there is a significant drop in sea surface temperature (SST), spanning large areas in the North Atlantic (Fig. A8a). This cooling

extends into fall and affects the surface air temperature accordingly. Conversely, when the eruptions start in September instead of June, there is no high-latitude SST decrease counteracting the LW trapping effects, and hence the warming signal. Here we have an example of how long-lasting effusive eruptions can lead to cumulative effects. This prolonged cooling signal into fall from eruptions starting in June appears for all scaling factors considered in this study and increases in magnitude with larger 260 eruptions (not shown here).

The focus of this study is on the instantaneous climate response to volcanic eruptions as a result of interactions between aerosols, clouds, and radiation, but we also see emerging dynamical effects in our simulations. In addition to the SST effect discussed earlier, the model shows atmospheric circulation changes. Most notably, we find a deepening of the Icelandic subpolar low during winter and weakening during summer (Fig. A11), resulting in a higher North Atlantic Oscillation (NAO) index in 265 winter and lower in summer (Fig. A9).

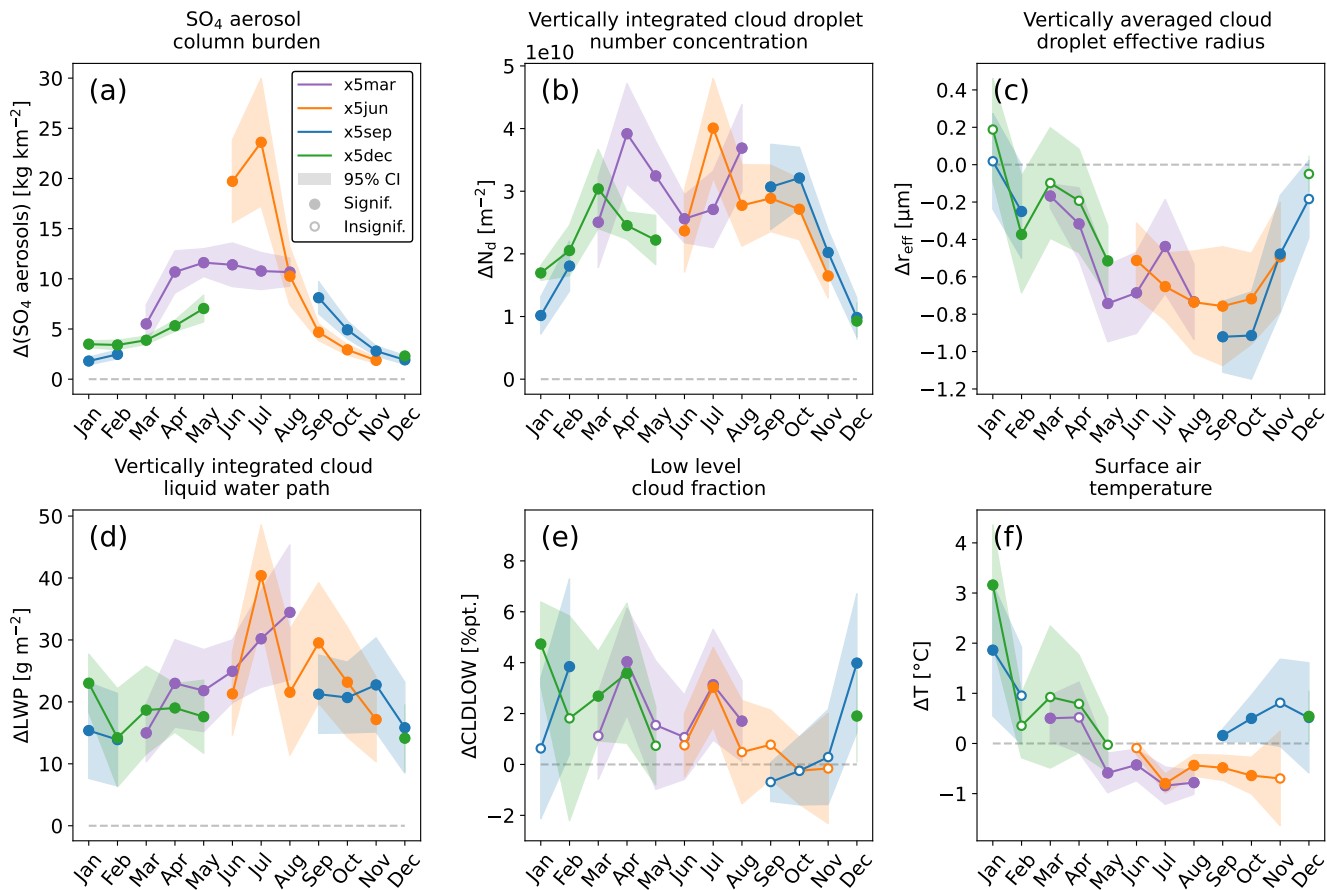

**Figure 6.** Monthly Arctic (as defined by the Arctic circle) mean anomalies for four different eruption scenarios, starting in March (`x5mar`), June (`x5jun`), September (`x5sep`), and December (`x5dec`): (a) $SO_4$ aerosol column burden, (b) vertically integrated cloud droplet number concentration, (c) vertically averaged cloud droplet effective radius, (d) vertically integrated cloud liquid water path, (e) low level cloud fraction, and (f) surface air temperature. Shades indicate 95 % confidence intervals based on a two-tailed $t$-test. Filled dots indicate anomalies significantly different from zero, open dots insignificantly.

### 3.7 Eruption size

So far we have discussed eruptions about 5 times the size of the 2014-15 Holuhraun eruption. Now, we include three additional scaling factors: $\times 1$, $\times 25$, and $\times 50$.

Fig. 7 shows mean anomalies north of the Arctic circle for the first three months of the eruption as a function of eruption size.
The $SO_4$ aerosol column burden anomalies scale almost linearly with the $SO_2$ emission strength, both in summer and winter. Since two of the three oxidants responsible for the oxidation of $SO_2$ in CESM2(CAM6)'s chemistry scheme are prescribed, namely OH and ozone, these oxidants will not get depleted over longer periods of time. Instead they are replenished at each

model timestep. This might lead to an overestimate of $SO_4$ production for the largest eruptions in our simulations. However, similar sulfur chemistry schemes with prescribed oxidants have been used in previous modelling studies investigating aerosols

and aerosol-cloud interactions without identifying such issues (e.g., Gettelman et al., 2015; Malavelle et al., 2017; Karset et al., 2018). It is known that stratospheric oxidants get depleted in the plumes of large explosive eruptions, leading to a slower oxidation rate of $SO_2$ with greater $SO_2$ emissions, hence a non-linear $SO_4$ aerosol formation in the stratosphere (Pinto et al., 1989; Bekki, 1995; Savarino et al., 2003; Case et al., 2023). This provides a motivation for future studies to explore such constraints in tropospheric volcanic plumes rising from large effusive eruptions.

The anomalies of other key variables do not show this linear behaviour but rather level out with eruption size, indicating that clouds become less sensitive to CCN perturbations at higher CCN levels. This saturation effect is well established and expected (e.g., Bellouin et al., 2020; Wang et al., 2024).

Previously in this study we have discussed how clouds become opaque to LW radiation when the LWP exceeds about 30 g m$^{-2}$, hence placing an upper limit on their LW trapping abilities. This is highlighted in Fig. 7f, where the model shows no

statistical difference between the winter temperature anomalies in the $\times 5$, $\times 25$, and $\times 50$ scaling scenarios. In the case of the summer cooling, a plateau seems to be reached at much higher emissions, with the $\times 25$ and $\times 50$ scaling scenarios yielding a significantly stronger cooling than the $\times 1$ and $\times 5$ scenarios. Fig. 7f also shows how the ensemble members better agree on the exact magnitude of the summer cooling than the winter warming, highlighting the role of the large meteorological variability during winter in the Arctic. The spring and fall anomalies mostly lie between the ones in summer and winter (Fig. A10). The

size of an effusive eruption, therefore, strongly influences the climate response.

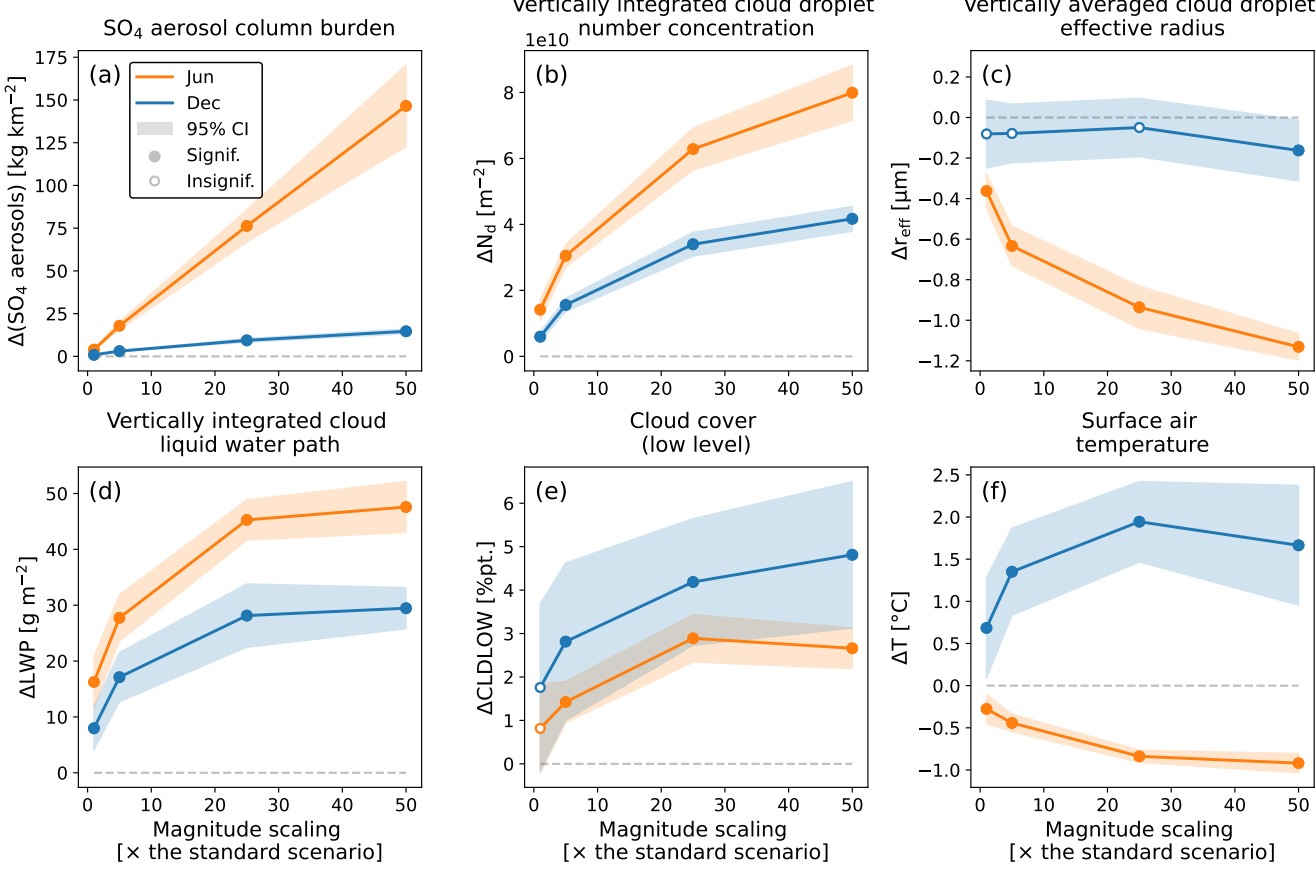

**Figure 7.** Mean anomalies for the first three months of the eruption north of the Arctic circle for four different eruption scaling scenarios ($\times 1$, $\times 5$, $\times 25$, $\times 50$) for: (a) SO$_4$ aerosol column burden, (b) vertically integrated cloud droplet number concentration, (c) vertically averaged cloud droplet effective radius, (d) vertically integrated cloud liquid water path, (e) low level cloud cover, and (f) surface air temperature. Dots indicate ensemble means and shades 95 % confidence intervals based on a two-tailed $t$-test. Orange represents eruptions starting in June and blue eruptions starting in December. Filled dots indicate anomalies significantly different from zero, open dots insignificantly.

## 4  Discussions

### 4.1  Extrapolation of the model simulations

Our main goal with this study is to explore the climate response to high-latitude, effusive volcanic eruptions as a function of eruption season and size. Producing a high-frequency dataset, for example by including more densely spaced magnitude scaling factors, is not viable due to the high computational cost of running an Earth system model. However, by extrapolating the model output, we can gain insight into what happens between our simulated scenarios. One such extrapolation is to fit the

data in Fig. 7 with the logarithmic curve described in Eq. 4. Table 1 shows the resulting values of the fitting coefficients $a$ and $b$.

**Table 1.** The fitting coefficient $a$ and $b$ for Eq. 4 for the variables in Fig. 7, excluding $SO_4$ aerosols which have a nearly linear relationship with the volcanic $SO_2$ emissions. Summer is the June to August mean for eruptions starting in June, winter is the December to February mean for eruptions starting in December. The unit for the fitting coefficient $a$ is the same as the unit for the fitted variable, and $b$ is dimensionless.

| | | $N_d$ | $r_{eff}$ | LWP | CLD-LOW | T |
|---|---|---|---|---|---|---|
| | | $[m^{-2}]$ | $[\mu m]$ | $[g\,m^{-2}]$ | $[\%]$ | $[°C]$ |
| Summer | $a$: | $2.27 \times 10^{10}$ | -0.204 | 8.86 | 0.588 | -0.194 |
| (JJA) | $b$: | 0.628 | 4.51 | 5.04 | 2.79 | 2.41 |
| Winter | $a$: | $1.28 \times 10^{10}$ | -0.012 | 6.13 | 0.814 | 0.286 |
| (DJF) | $b$: | 0.512 | 259 | 3.00 | 6.96 | 15.4 |

As discussed in Section 3.7, most anomalies gradually level off as the eruptions get larger. In other words, the magnitude of the climate response is sensitive to variations in eruption size for small eruptions but insensitive for large eruptions. To get a measure for when this plateau is reached, we define a threshold for the growth rate in Eq. 5. Here we choose a threshold value of 1 % increase in absolute anomalies per scaling factor. This is a small, arbitrary number, meant to indicate when the growth rate starts to level of, and should rather be viewed as guiding value than a hard separator. We consider eruptions resulting in a growth rate above this threshold to be in the "sensitive stage" (those are smaller eruptions) but eruptions resulting in a growth rate below it to have reached the "plateau stage" (those are larger eruptions). Table 2 shows the magnitude scaling factors corresponding to the 1 % threshold. When comparing summer and winter, the 1 % threshold is generally reached for similarly sized eruptions. The surface air temperature is an exception where the growth rate decreases much faster in winter. As for the cloud droplet effective radius during winter, the logarithmic fit does not offer much information since the mean Arctic anomalies remain constant as a function of eruption size. In most cases, the 1 % threshold is reached for scaling factors between $\times 20$ and $\times 30$. We would, therefore, expect the magnitude of the climate response to be more sensitive to the size of the eruption for eruptions smaller than about 20 times the size of the Holuhraun eruption and less sensitive to the eruption size for eruptions larger than about 30 times. This applies for both summer and winter. The largest known effusive eruptions in Iceland were most likely around 20 times larger than the 2014-15 Holuhraun eruption (see Sect. 1) and we would therefore expect them to either have reached or been close to reaching the "plateau stage".

**Table 2.** Magnitude scaling factors when the growth rate from Eq. 5 drops below 1 % per scaling factor for the logarithmic fits (Eq. 4, fitting coefficients from Table 1) of the variables in Fig. 7, excluding $SO_4$ aerosols.

| | $N_d$ | $r_{eff}$ | LWP | CLD-LOW | T |
|---|---|---|---|---|---|
| Summer (JJA) | $\times 31$ | $\times 22$ | $\times 21$ | $\times 23$ | $\times 24$ |
| Winter (DJF) | $\times 33$ | $\times 12$ | $\times 23$ | $\times 20$ | $\times 18$ |

## 4.2 The 21st century Fagradalsfjall fires

Within volcanology, the term *fires* refers to a single long-lasting volcanic eruption or a series of individual but connected eruptions. An example of the former is the 1784-85 Laki eruption (also known as the Skaftá fires) and an example of the latter are the 1975-1984 Krafla fires, both in Iceland. These fires typically last years (Thordarson and Larsen, 2007). In 2021, a series of eruptions started on the Reykjanes peninsula in Iceland. Collectively, these eruptions have not received an official name yet but they are often referred to as the Fagradalsfjall fires. As of this writing, these fires are still ongoing.

The eruptions in the Fagradalsfjall fires share many similarities with the eruptions simulated in this study. They have all been effusive, their eruption plumes have mostly stayed below 3 km above sea level, and they have lasted between a few days and several months. The first eruption in the series, 2021 Fagradalsfjall, has been the longest to date, lasting six months from March 19th to September 18th 2021 (Pfeffer et al., 2024). Coincidentally, the 2014-15 Holuhraun eruption also lasted six months but it started in fall (Gíslason et al., 2015). Holuhraun was, however, a much larger eruption, with estimated total $SO_2$ emissions of about 9.6 Tg (Pfeffer et al., 2018) compared to Fagradalsfjall's 0.97 Tg (Pfeffer et al., 2024). This gives the 2021 Fagradalsfjall eruption a magnitude scaling factor of about $\times 0.1$ within the framework of our study.

Table 3 lists estimated Arctic anomalies for a $\times 0.1$ sized eruption based on Eq. 4 and our simulations for the first three months of eruptions starting on the first days of March (spring), June (summer), September (fall), and December (winter). Of those, the spring eruption most closely resembles the Fagradalsfjall eruptions in terms of starting date. These estimated anomalies for $\times 0.1$ sized eruptions are very small and unlikely to stand out from natural variability. Other eruptions in the Fagradalsfjall fires have been much shorter, lasting only a few days or a few weeks (e.g., Esse et al., 2023; Sigmundsson et al., 2024), limiting their potential climate impacts due to the short lifetime of volcanic sulfur in the troposphere (e.g., Chin and Jacob, 1996; Schmidt and Carn, 2022). It is therefore unlikely that Fagradalsfjall fires have caused significant climate impacts in the Arctic so far.

**Table 3.** Estimated Arctic anomalies from a ×0.1 sized eruption using Eq. 4 and the simulations performed in this study. Spring refers to the March to May mean for an eruption starting in March, summer refers to the June to August mean from an eruption starting in June, fall refers to the September to November mean from an eruption starting in September, and winter to the December to February mean from an eruption starting in December. The numbers in the parenthesis are control means.

| | $\Delta N_d$ [m$^{-2}$] | $\Delta r_{eff}$ [$\mu$m] | $\Delta$LWP [g m$^{-2}$] | $\Delta$CLD-LOW [%pt.] | $\Delta$T [°C] |
|---|---|---|---|---|---|
| Spring (MAM) | $22 \times 10^8$ ($132 \times 10^8$) | -0.1 (2.7) | 4 (58) | 0.4 (75.6) | ~0.0 (-12.1) |
| Summer (JJA) | $14 \times 10^8$ ($271 \times 10^8$) | -0.1 (5.7) | 4 (144) | 0.1 (84.7) | ~0.0 (2.9) |
| Fall (SON) | $18 \times 10^8$ ($126 \times 10^8$) | -0.1 (4.4) | 5 (104) | ~0.0 (86.2) | 0.5 (-5.4) |
| Winter (DJF) | $6 \times 10^8$ ($49 \times 10^8$) | ~0.0 (1.9) | 2 (35) | 0.4 (71.3) | 0.3 (-23.0) |

## 4.3 Observational evidence

The fall of 2014 was warm over the Greenland Sea (e.g., November in Ittoqqortoormiit, Jan Mayen, and Grímsey, Fig. A12b). Through observational and modelling evidence, Zoëga et al. (2023) argue that the 2014-15 Holuhraun eruption contributed to this warming signal through increased cloud LW trapping under limited sunlight. Although the simulations performed for this study here are not designed to exactly reproduce the 2014-15 Holuhraun eruption (for example with respect to the meteorology at the time) we do, nevertheless, see similarities in the climate response. When comparing anomalies from our `x1sep` simulations (which very closely resemble the 2014-15 Holuhraun eruption in terms of emissions and timing) averaged over the Greenland Sea (approximated by the area between 65°N to 80°N, and 25°W to 5°E) to anomalies from the ERA5 reanalysis for the same area, we find a warming signal in the fall months of September to November in both cases (Fig. A12a). This, along with the results of Zoëga et al. (2023), lends support to the credibility of the high-latitude winter warming mechanism discussed here.

## 4.4 Model dependencies

Aerosol-cloud interactions are among the largest sources of uncertainty in our understanding of the climate system, both from observational and modelling perspectives. This holds especially true for LWP and cloud fraction adjustments to aerosol perturbations (Forster et al., 2021). In a previous study, Malavelle et al. (2017) compared the cloud response to aerosol perturbations

from the 2014-15 Holuhraun eruptions from several different climate models. Among them was CAM5, the predecessor of CAM6, which they found to produce an overly strong LWP response over the open ocean areas around Iceland compared to satellite retrievals. This result is further supported by a modelling study by Haghighatnasab et al. (2022), which found neither LWP nor cloud cover response attributable to that eruption. In contrast, recent studies using machine learning to analyse satellite data have found that the Holuhraun eruption did indeed lead to a significant increase in cloud fraction (Chen et al., 2022; Wang et al., 2024). Furthermore, analyses of both observational data (Zhao and Garrett, 2015) and satellite retrievals (Murray-Watson and Gryspeerdt, 2022) have found a positive relationship between LWP and cloud droplet number concentration in the Arctic, which is where our simulations show the strongest increase in cloud LW trapping. The excessive LWP response in CAM5 reported by Malavelle et al. (2017) has since been addressed in CAM6 by modifications of the aerosol-cloud interaction processes, making the LWP less sensitive to perturbations in the cloud droplet number concentration (Gettelman and Morrison, 2015; Danabasoglu et al., 2020).

## 5    Conclusions

In this study, we use the Earth system model CESM2(CAM6) to systematically investigate the climate impacts of northern hemisphere, high-latitude, long-lasting effusive volcanic eruptions (similar to the 2014-15 Holuhraun eruption in Iceland) as a function of eruption season and size. This systematic approach provides us with a broad view of the climate impacts of such eruptions and allows us to make quick estimates of the climate impacts of a wide range of effusive volcanic eruptions in Iceland. Our main results are twofold:

- The climate response to high-latitude effusive volcanic eruptions is strongly modulated by different seasons. For winter eruptions the model shows surface warming in the Arctic and for summer eruptions it shows surface cooling at mid-latitudes and in the Arctic. The main contributors to this seasonal dependency are the availability of sunlight and atmospheric oxidants, the Arctic sea ice cover, and the background CCN and low level cloud states.

- As eruptions get larger in terms of $SO_2$ emissions, the magnitude of the climate response becomes less sensitive to variations in eruption size. In other words, the rate of change of the climate response as a function of eruption size is non-linear and decreases with increased eruption size. For eruptions below ca. 20 to 30 times the size of the 2014-15 Holuhraun eruption, the magnitude of the climate response is highly sensitive to the size of the eruption. For larger eruptions, the climate response becomes saturated, displaying minor variations with increased $SO_2$ emissions.

When the climate impacts of effusive volcanic eruptions are discussed, the focus is usually on their cooling effects as a result of increased reflectance of sunlight (e.g., Eguchi et al., 2011; Schmidt et al., 2012; Malavelle et al., 2017). We do, however, have evidence for the opposite, namely a significant warming in the Arctic in the early winter as a result of a long-lasting, effusive volcanic eruption (Zoëga et al., 2023). In this study, we have illustrated how sensitive the climate response to such eruptions is to the season of the eruption and how a surface warming is the dominant response at high latitudes during winter.

That effusive volcanic eruptions lead to surface cooling is therefore an oversimplification according to our results, especially in the Arctic.

In light of the high effusive volcanic activity in Iceland, especially during the past decade (e.g., 2014-15 Holuhraun and the ongoing Fagradalsfjall fires on the Reykjanes peninsula), the potential for very large eruptions (e.g., 1783-84 Laki and 939-940 Eldgjá), the rapidly changing climate in the Arctic, and the similarities to cloud seeding geoengineering, understanding the climate impacts of high-latitude effusive volcanic eruptions becomes increasingly relevant.

*Code and data availability.* The relevant model output data underlying the figures of this manuscript will be freely available online at time of publication along with a Jupyter notebook containing plotting scripts. The ERA5 reanalysis is available at the Copernicus Climate Change Service (C3S) Climate Data Store (CDS) (Hersbach et al., 2024). Observational timeseries are available at NCCS (2023) for Svalbard lufthavn/airport and Jan Mayen, DMI (2023) for Danmarkshavn and Ittoqqortoormiit, and Icelandic Met Office (2024) for Grímsey.

## Appendix A

### A1  Background aerosol and cloud conditions

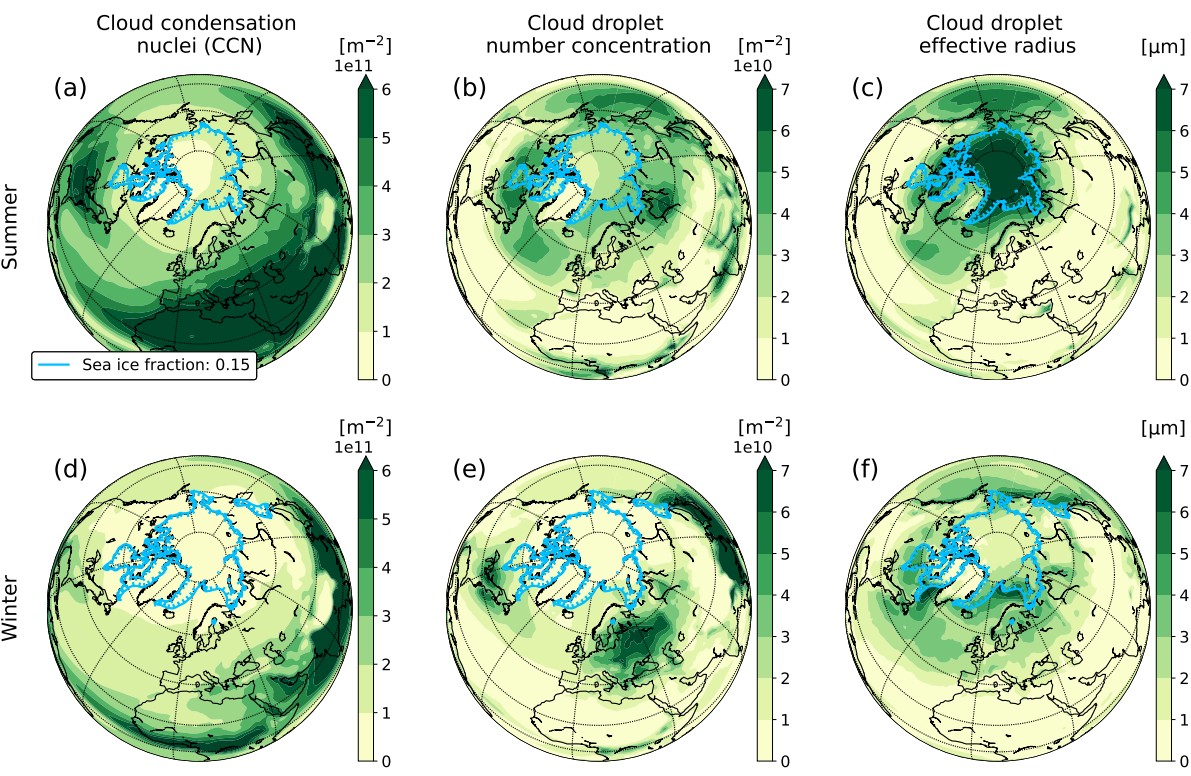

**Figure A1.** Summer (June to August) and winter (December to February) means from the CESM2(CAM6) control run for cloud condensation nuclei (a and d), cloud droplet number concentration (b and e), and cloud droplet effective radius (c and f).

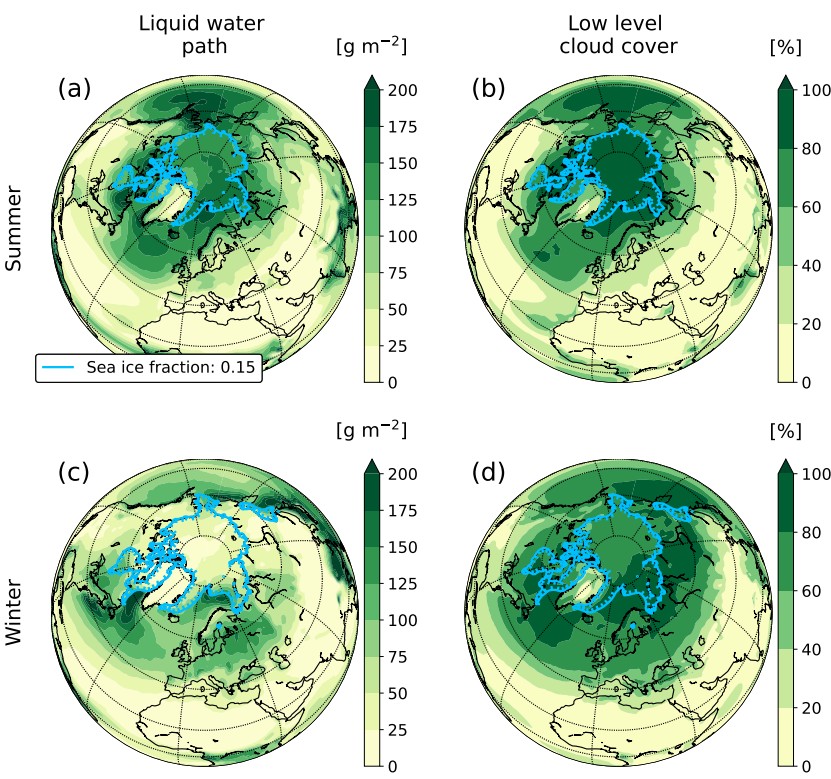

**Figure A2.** Same as Fig. A1 but for liquid water path (a and c) and low level cloud cover (b and d).

## A2 Precipitation

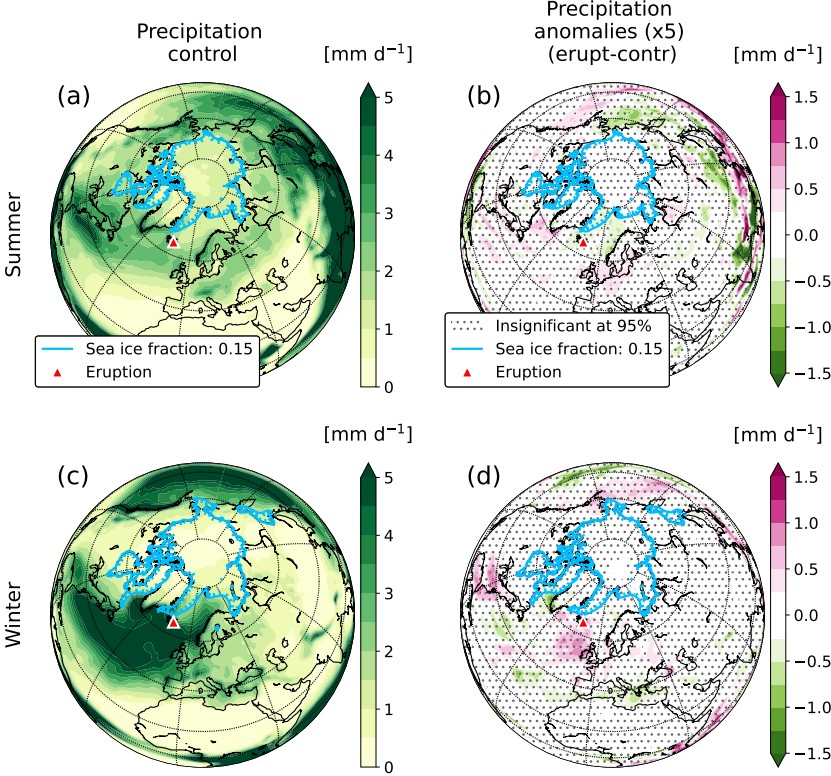

**Figure A3.** Precipitation from the CESM2(CAM6) simulations. Control means for (a) summer (June to August) and (c) winter (December to February). Mean anomalies for the first three months of the eruption for (b) `x5jun` and (d) `x5dec`.

**Figure A4.** Vertical profiles for mean summer (June to August) and winter (December to February) background conditions from the control run in the top row and mean anomalies for the first three months of the `x5jun` and `x5dec` eruption scenarios in the bottom row. Means over the Arctic Sea ice bounded by 75°N-90°N and 20°W-160°W. Cloud condensation nuclei (a and e), cloud droplet number concentration (b and f), cloud droplet effective radius (c and g), and liquid water content (d and h).

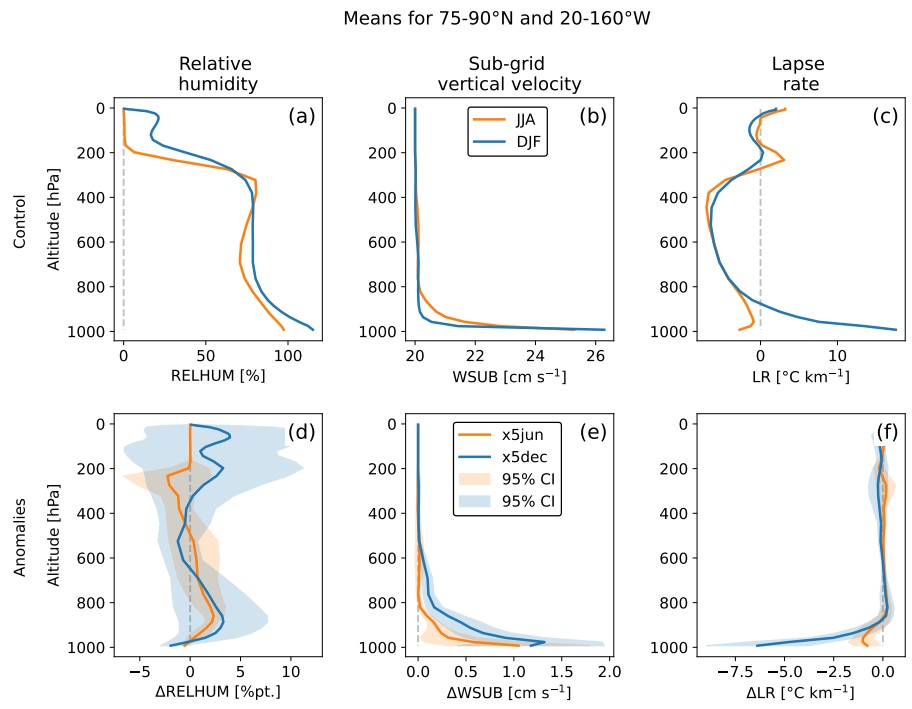

**Figure A5.** Same as Fig. A4 but for relative humidity (a and d), sub-grid vertical velocity (b and e), and lapse rate (c and f).

## A4  Direct aerosol effects

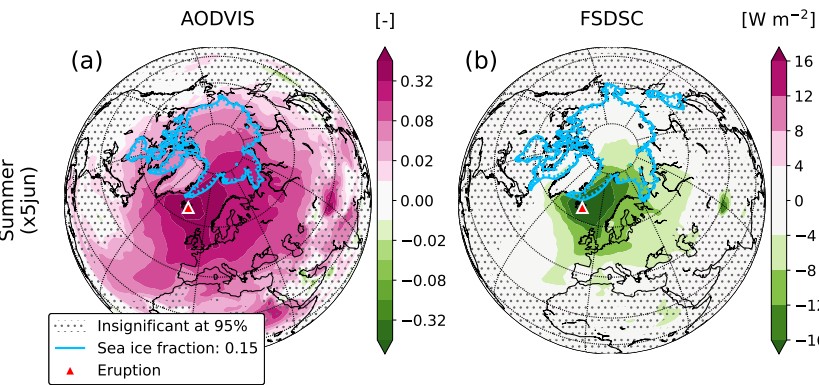

**Figure A6.** Mean anomalies for the first three months of the eruption for the `x5jun` scenario for (a) the aerosol optical depth at 550 nm (AODVIS) and (b) downward clear-sky SW flux at the surface (FSDSC).

## A5 Surface albedo

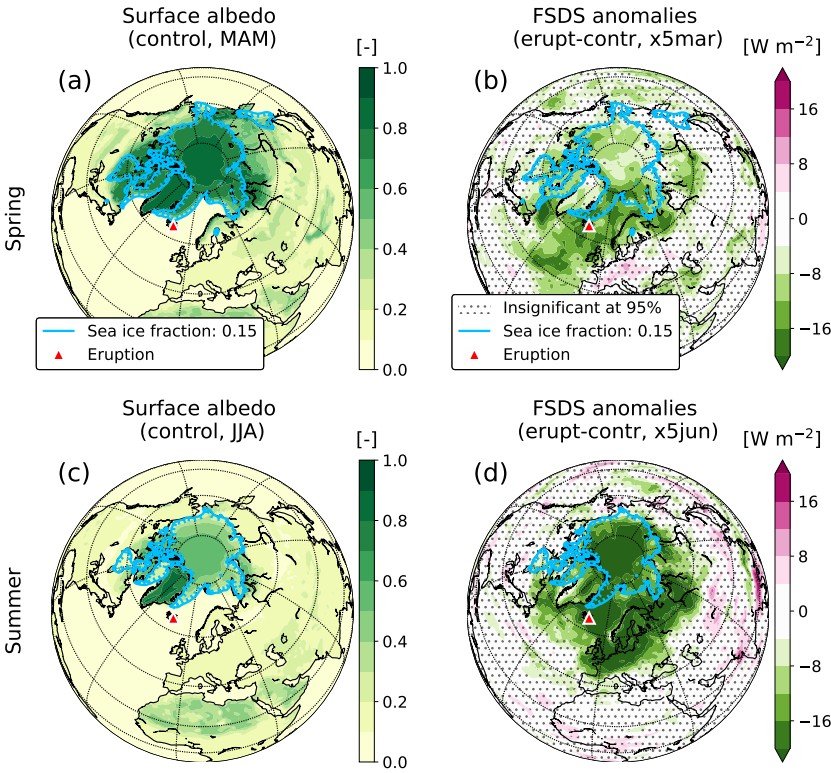

**Figure A7.** Surface albedo means from the control run for (a) spring (March to May), and (c) summer (June to August). Mean net surface downward shortwave radiation (FSDS) anomalies for the first three months of the (b) `x5mar`, and (b) `x5jun` simulations.

## A6    Sea surface temperature and sea ice cover

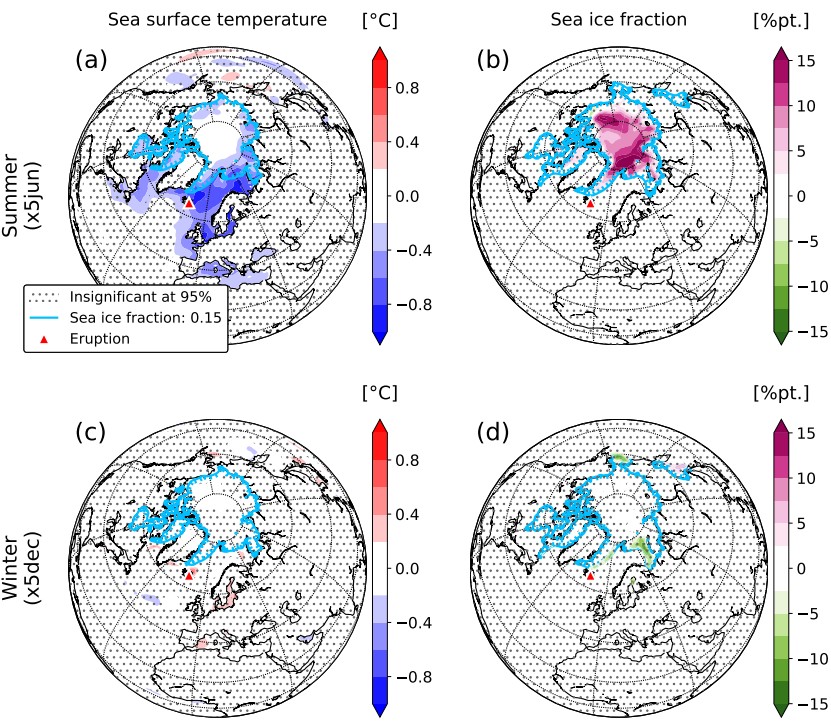

**Figure A8.** Same as Fig. 2 but for sea surface temperature (a and c), and sea ice fraction (b and d).

## A7    The North Atlantic Oscillation

We calculate the North Atlantic Oscillation (NAO) index from our model data as the difference in normalized sea level pressure between the Azores (38.2°N, 27.0°W) and Stykkishólmur in Iceland (65.1°N, 22.7°W). That is,

$$\mathrm{NAO_{ind}} = P'_{\mathrm{Az}} - P'_{\mathrm{St}} \tag{A1}$$

where $P'_{\mathrm{Az}}$ and $P'_{\mathrm{St}}$ are normalized sea level pressures for the Azores and Stykkishólmur respectively, and

$$P' = \frac{P - \bar{P}}{\sigma_P} \tag{A2}$$

where $\bar{P}$ and $\sigma_P$ are the mean sea level pressure and standard deviation from the control run respectively.

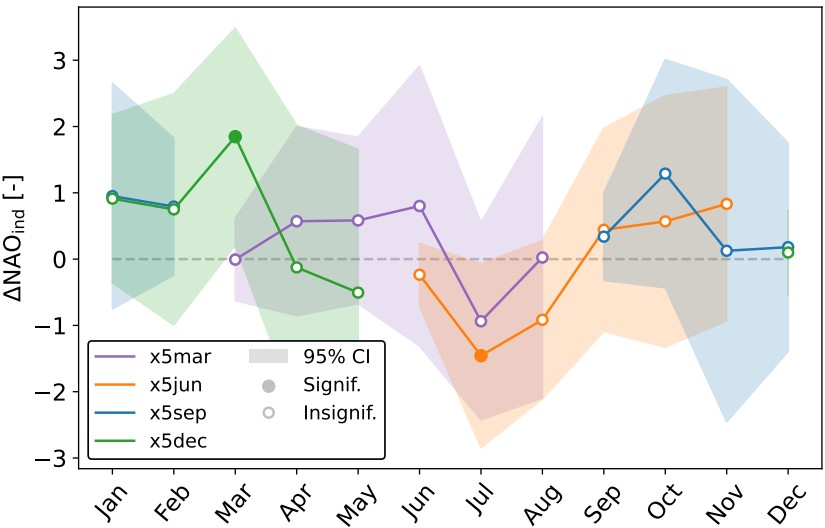

**Figure A9.** Modelled monthly mean North Atlantic Oscillation (NAO) index anomalies for eruptions using the ×5 scaling factor. The NAO index is calculated as the difference in normalized sea level pressure between the Azores and Stykkishólmur in Iceland.

## A8    Spring and fall

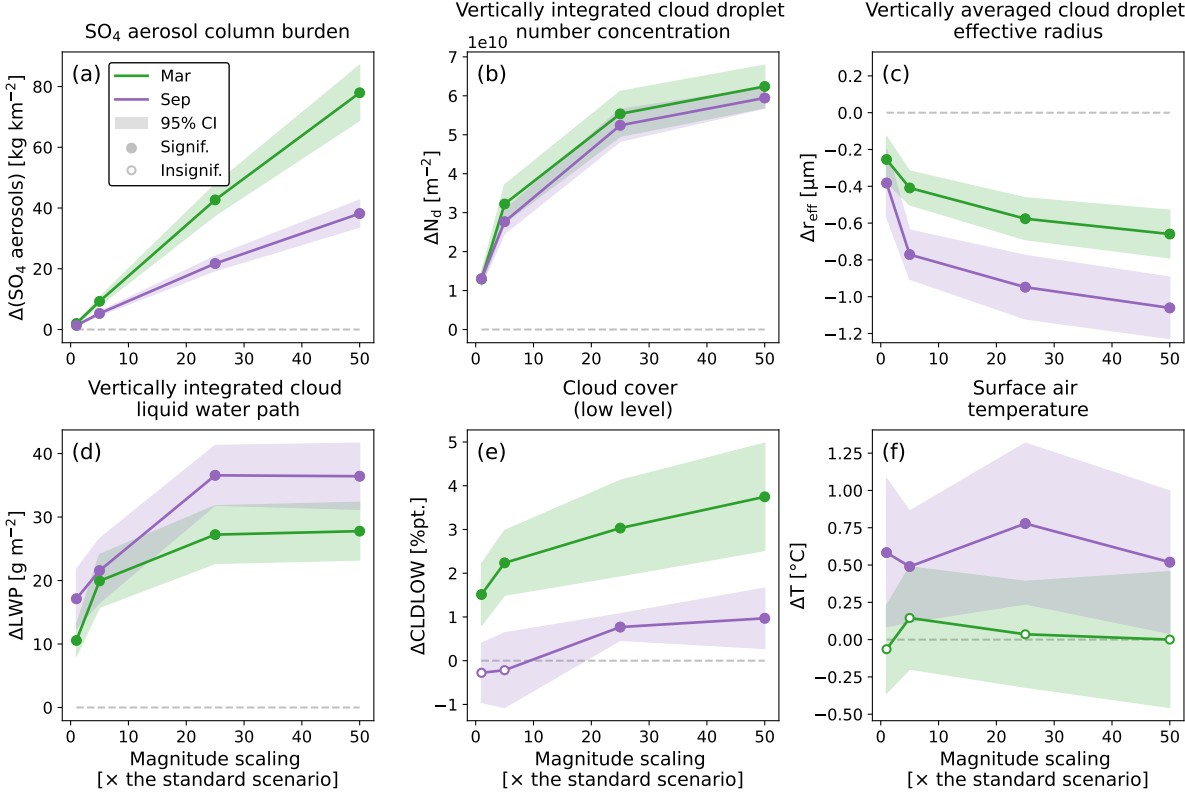

**Figure A10.** Same as Fig. 7 but for eruptions starting in March (green) and September (purple).

**Figure A11.** Sea level pressure anomalies for the first three months of an eruption for the (a) `x5mar`, (b) `x5jun`, (c) `x5sep`, and (d) `x5dec` scenarios. Grey contours are control means.

## A10  Observational evidence: Comparing the 2014-15 Holuhraun eruption and the `x1sep` simulations

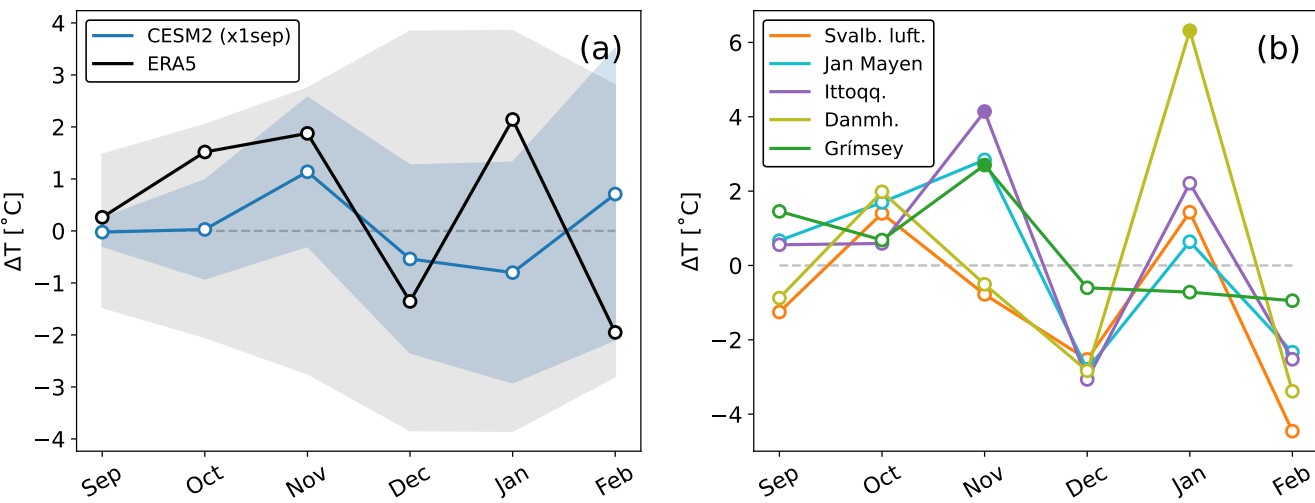

**Figure A12.** (a) Monthly mean surface air temperature anomalies averaged over the Greenland Sea as defined by the area between 66°N to 80°N, and 25°W to 5°E. The blue line shows the `x1sep` CESM2(CAM6) ensemble mean anomalies. The black line shows anomalies in September 2014 to February 2015 relative to the 1984 to 2013 (30 years) climatology from the ERA5 reanalysis. Shades indicate 95 % confidence interval and 95 % prediction interval for the CESM2(CAM6) simulations and ERA5 respectively. (b) Observed monthly mean surface air temperature anomalies from stations in and around the Greenland Sea: Svalbard lufthavn/airport (78.25°N, 15.50°E; orange), Jan Mayen (70.94°N, 8.67°W; cyan), Ittoqqortoormiit (70.48°N, 21.95°W; purple), Danmarkshavn (76.77°N, 18.68°W; olive), and Grímsey (66.54°N, 18.02°W; green). Anomalies are for September 2014 to February 2015 relative to the 1984 to 2013 climatology. Prediction intervals are not plotted to preserve clarity. For all timeseries in both (a) and (b), filled dots indicate significant anomalies, open dots insignificant.

*Author contributions.* TZ, TS, and KK conceived the study. TZ performed the model simulations and data analysis. TZ led the manuscript writing with input from TS and KK.

*Competing interests.* The authors declare no competing interests.

*Acknowledgements.* This project received funding from the European Union's Horizon 2020 research and innovation program under the Marie Skłodowska-Curie grant agreement No. 945371 through the "CompSci: Training in Computational Science" doctoral program launched and managed by the Faculty of Mathematics and Natural Sciences at the University of Oslo. TS would additionally like to acknowledge funding from the European Union's Horizon Europe program under the ERC Consolidator grant agreement No. 101045273. KK would

additionally like to acknowledge funding from the Research Council of Norway/University of Oslo Toppforsk project "VIKINGS" with the grant no. 275191. The simulations in this study were performed on the Fram high performance computer and the model output stored on the National Infrastructure for Research Data (NIRD), both provided by Sigma2 and the Norwegian Research Infrastructure Services (NRIS) in Norway.

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
