# Peer review of "Modelled surface climate response to Icelandic effusive volcanic eruptions: Sensitivity to season and size"

_EGUsphere, 2024_

## Author Response (AR1)

**Author's response**

We thank the two anonymous reviewers and the editor for their detailed feedback on our manuscript. Below we include point-by-point replies to the reviews. The comments from the reviewers are written in black, our replies in orange.

**1 Anonymous Referee 1**

Zoëla et al. are interested in the climate impacts of effusive volcanic emissions of sulfur via their interactions with radiation and with clouds. They ask how the effect depends on magnitude of sulfur emission rate and on seasonal timing of the emission. The study makes use of an atmospheric general circulation model. The study in general is well written and of interest to the readership of Atmos. Chem. Phys.

I have two main comments and a number of specific ones.

**1.1 Main comment 1**

Model-dependency of the results. This is a model-only study. The qualitative results are not very surprising, and so it is the quantitative results that matter. Most of the relevant results (cloud response, related radiation response especially in the terrestrial spectrum, and subsequently, temperature and sea ice extent responses) hinge on the simulated response of cloud liquid and ice water path and cloud horizontal cover to the emitted sulfur. The model the authors use (CAM6) shows a strongly positive response of cloud liquid water path and cloud cover to increasing aerosols. In the study of Malavelle et al. (Nature 2017, cited), a number of general circulation models were evaluated for their response to Holuhraun aerosol. The predecessor to the model this study uses, CAM5, was one of these models. CAM5 showed, erroneously, a strong positive LWP response. The data, and some other models, showed no net response of LWP. This result has been confirmed since then from other model studies (e.g. Haghighatnasab et al. ACP 2022). So the authors should at least discuss how peculiarities of the model they use, especially biases, affect the outcome.

- We have added a new section to the manuscript to address this comment. It is section "4.2 Model dependencies" and reads as follows:

  "*Aerosol-cloud interactions are among the largest sources of uncertainty in our understanding of the climate system, both from observational and modelling perspectives. This holds especially true for LWP and cloud fraction adjustments to aerosol perturbations (Forster et al., 2021). In a previous study, Malavelle et al. (2017) compared the cloud response to aerosol perturbations from the 2014-15 Holuhraun eruptions from several different climate models. Among them was CAM5, the predecessor of CAM6, which they found to produce an overly strong LWP response over the open ocean areas around Iceland compared to satellite retrievals. This result is further supported by a modelling study by Haghighatnasab et al. (2022), which found neither LWP nor cloud cover response attributable to that eruption. In contrast, recent studies using machine learning to analyse satellite data have found that the Holuhraun eruption did indeed lead to a significant increase in cloud*

*fraction (Chen et al., 2022; Wang et al., 2024). Furthermore, analyses of both observational data (Zhao and Garrett, 2015) and satellite retrievals (Murray-Watson and Gryspeerdt, 2022) have found a positive relationship between LWP and cloud droplet number concentration in the Arctic, which is where our simulations show the strongest increase in cloud LW trapping. The excessive LWP response in CAM5 reported by Malavelle et al. (2017) has since been addressed in CAM6 by modifications of the aerosol-cloud interaction processes, making the LWP less sensitive to perturbations in the cloud droplet number concentration (Gettelman and Morrison, 2015; Danabasoglu et al., 2020)."*

**1.2   Main comment 2**

Detection attempts. One of the simulations the authors perform (September) is starting at the time the effusion from Holuhraun in 2014 indeed started. It would be useful if the authors in particular discussed this simulation and discussed in particular to which extent the outcomes they simulate might be attributable in reality.

- The effusive 2014-15 Holuhraun eruption is in fact a topic of our previous study (see Zoëga et al., 2023, https://doi.org/10.21203/rs.3.rs-3249183/v1), which is under review. In that study, we analyse observational and reanalysis data, along with performing simulations using CESM2(CAM6), and find that the 2014-15 Holuhraun eruption contributed to surface warming over the Greenland and Norwegian Seas in the fall of 2014 through the same processes as described in this current manuscript. We address this in the last sentence of the second paragraph in the Introduction (section 1 in the main manuscript).

**1.3   Minor remarks**

- l14 – this statement depends on what is considered "high". There are also many small emission sources.

  – *"They release high amounts of gases ..."* has been changed to *"They release various gaseous species ..."* There is a more detailed discussion about the magnitude of $SO_2$ emissions from effusive eruptions later in the introduction.

- l19 – this is an old-fashioned term (actually both), typically one nowadays would talk about "radiative forcing due to aerosol-cloud interactions". Also it is the drop number enhancement that leads to a higher albedo (if smaller droplets are mentioned, one needs to say something about the cloud liquid water path)

  – *"Previous studies have observed the so-called first indirect aerosol effect, or the cloud albedo effect (decreased cloud droplet size and higher cloud albedo with increased CCN concentrations) ..."*

    has been changed to

    *"Previous studies have observed a shortwave radiative forcing due to aerosol-cloud interactions (more numerous cloud droplets and higher cloud albedo with increased CCN concentrations) ..."*

- l23 – this is an outdated term, nowadays one would describe it as "adjustments to aerosol-cloud interactions". The cloud lifetime hypothesis is only one of many.

- "*The second indirect aerosol effect, or the cloud lifetime effect (changes in cloud cover and cloud liquid water path with increased CCN concentrations and smaller cloud droplets) (Albrecht, 1989), has ...*"

  has been changed to

  "*Adjustments to aerosol-cloud interactions (Albrecht, 1989) have ...*"

- l32 – previously, the authors have described effusive emissions in contrast to eruptions, why is now effusion just a variant of explosion?

  - One way of classifying volcanic eruptions is to place them on a scale from effusive to explosive. In our study, we use those terms, that is, "effusive" and "explosive", to refer to eruptions on each end of this spectrum. A single eruption can have both effusive and explosive characteristics. We refer to those eruptions as "mixed effusive-explosive". Throughout the study we use the terms "effusive eruptions" or "effusive volcanic eruptions". In order to clarify the terminology, we have exchanged the first sentence in the Introduction ("*Effusive volcanic eruptions are characterised by non-explosive activity*") with the following text:

    "*Volcanic eruptions vary greatly in their behaviour. Some are dominated by explosive activity where the magma explodes and is erupted as tephra. In other cases, explosive activity is mostly absent and the magma is mainly erupted as lava. Eruptions falling into the latter group are referred to as effusive eruptions.*"

- l59 – more details on the aerosol scheme are required. What types are considered, are they internally mixed, are conversion processes simulated?

  - The following text has been added to the model description in section 2.1:

    "*The four log-normal aerosol modes of MAM4 are Aitken, accumulation, coarse, and primary carbon. Together they include sulfate, sea salt, primary and secondary particulate organic matter, black carbon, and soil dust, which are internally mixed within each mode. The conversion of aerosol from one mode to another is simulated through coagulation and condensation (Liu et al., 2012, 2016).*"

- l64 – the CMIP6 historical forcing stops in 2014, what is the assumption for 2015?

  - In order to clarify which forcing data we use for the last five months of our control run (January to May 2015), we have added the following text to section 2.2 in the manuscript:

    "*For the year 2015, extensions of the existing historical CMIP6 forcing fields were used when available (van Marle et al., 2017; Hoesly et al., 2018), otherwise the SSP2-4.5 forcing (O'Neill et al., 2016) was applied.*"

- l76 – the authors earlier explained that the largest-ever eruption had 21 m$^3$ lava, Holuhraun 1 m$^3$ So a factor of 50 seems not plausible. It may be interesting, of course, nevertheless

  - This is correct. A factor of 50 is not plausible for Icelandic effusive eruptions. We have replaced the sentence

*"In addition to the ×1 scaling factor, corresponding to a Holuhraun-sized eruption, we perform simulations using scaling factors of ×5, ×25, and ×50, covering a plausible size range of Icelandic effusive eruptions."*

with

*"In addition to the ×1 scaling factor, corresponding to a Holuhraun-sized eruption, we perform simulations using scaling factors of ×5 and ×25, covering the plausible range of Icelandic effusive eruptions, and ×50, extending into the size range of the largest known flood basalts on Earth (Kasbohm and Schoene, 2018)."*

- l115 – CCN?

    – Indeed!

- l146 – what is the energetic argument for the precipitation reduction? is a significant cooling of the atmosphere seen?

    – Over the eastern Labrador Sea (along the south-western coast of Greenland), where we model the strongest wintertime precipitation reduction (Fig. A3d), we also model surface air cooling (Fig. 5f). We also model slight precipitation decrease over the Sea of Okhotsk and the Kamchatka peninsula, accompanied by a cooling signal. To address this, we add the following sentence to the third paragraph of section 3.3:

    *"… accompanied by a cooling signal (Fig. 5f)."*

- l155 – this can be tested, since it is model-world only. Does the model precipitate more efficiently via ice phases?

    – During winter in the Arctic, the model shows an increase in the liquid to ice water content ratio during an eruption (ice water anomalies not shown in the manuscript). To get this better across we have replaced:

    *"It is well-established that when clouds contain more liquid and less ice, their lifetimes increase (e.g., Storelvmo et al., 2011), hence the increased cloud cover and LWP in the wintertime Arctic basin."*

    with

    *"It is well-established that when the ratio of liquid to ice water content in clouds increases, they generally precipitate less efficiently (the Wegener-Bergeron-Findeisen process) and their lifetimes increase (e.g., Tsushima et al., 2006; Storelvmo et al., 2011; Tan and Storelvmo, 2019), hence the increased cloud cover and LWP in the wintertime Arctic basin. This process is represented in CESM2(CAM6)."*

- l185 – why only "dominate"? during polar night, aerosol-radiation interactions are zero

    – Indeed. But the polar night only covers a part of the Arctic (as defined as the area north of the Arctic circle) part of the winter. Since there is some sunlight, albeit very little, we would like to use non-exclusive wording. That is, "indirect aerosol effects dominate" instead of "only indirect aerosol effects" or "aerosol-radiation interactions are absent".

- l192 – is this not potentially testable? even one-fifth of this signal could be detectable. Was this the case in reality?

  - See reply to Main comment #2.
  - Although effusive volcanic eruptions in the size range discussed in the manuscript have been common throughout Earth's history, the intervals between them can be quite long. For example, the 2014-15 Holuhraun eruption was the largest effusive eruption in Iceland since the 1783-84 Laki eruption (about ten times larger than Holuhraun in terms of mean SO2 emission rate), more than 230 years earlier. Smaller eruptions, for example 2021 Fagradalsfjall (about a tenth of the size of Holuhraun), are more common but, as discussed in our manuscript, they are unlikely to result in significant surface air temperature responses. Observational data on the climate impacts of large high-latitude, effusive volcanic eruptions is, therefore, scarce.

- l205 – same here, isn't this signal large enough to be detectable in reality?

  - See replies to Main comment #2 and Minor remark l192.

- l274 / Table 1 – the physical units for the a and b coefficients are missing

  - The following has been added to the caption of Table 1:

    "*The unit for the fitting coefficient a is the same as the unit for the fitted variable, and b is dimensionless.*"

- Throughout: References, where multiple publications are cited for one fact, in general should be ordered chronologically.

  - This has been taken into account. Now, all references are first ordered by year and then by the last name of the first author.

**2 Anonymous Referee 2**

This study investigates the aerosol-climate interactions following effusive volcanic eruptions that emit large amounts SO2. The study builds on previous analyses of aerosol-cloud interactions following the 2014-15 Holuhraun eruption, but adds a novel direction by simulating the climate impact if a similar eruption happened in different seasons or of increased magnitude of emissions. The results show how the climate response varies with the seasons of the eruption start – particularly there is opposing effects on temperature between summer and winter, and how the response plateaus for higher magnitude eruptions.

Overall, the study presents a nice piece of an analysis. The results are clearly presented and the article is well laid out. This study is an interesting addition to the literature on aerosol-climate interactions following volcanic eruptions. I recommend minor revision prior to publication.

**2.1 Comment 1**

Coupled simulations can evolve differently due to the variability of the climate. This study seems to use only use one initial condition ensemble member. How do you know that differences in cloud cover between the eruption vs control simulations could be just due to differences in climate variability or if the results might be different if another initial condition ensemble member is used? I'm not sure how much impact the variability in coupled simulations would have in the short timescale of this study but I think a bit of discussion/acknowledgment of this is needed since most other studies on this topic have used nudged atmosphere-only simulations.

- In our study we start with an 11 years long control run, corresponding to the model years 2005 to 2015 (see section 2.2). On the first days of March, June, September, and December we branch off simulations where we include volcanic emissions. This we do for the first ten years of the control run, resulting in ten eruption simulation for each combination of starting month and magnitude scaling. This is what we refer to as a ten member ensemble.

  For example, the ensemble of x5jun consists of ten branch simulations where the eruption is of size x5 and starts in June of each of the model years from 2005 to 2014. Each of the different Junes has its own meteorological state, different from the other Junes.

  We then perform a pairwise comparison between an eruption simulation and the corresponding part of the control run, isolating the impact of the volcanic eruptions under those specific conditions. Without an eruption, the branch simulation would be a bit-for-bit recreation of the control run. Hence the anomalies on which we base our analysis must be due to the volcanic eruptions. In order to clarify our experimental setup, we have divided section 2.2 into several paragraphs for better legibility and added the following sentence to the second paragraph of the revised manuscript:

  "*For each scenario considered in this study (see below), this leads to ten eruption simulations, each of which has its own unique initial conditions.*"

**2.2 Comment 2**

Most climate models don't represent the entrainment processes that would lead to a decrease in LWP for an increase in aerosol. There is no discussion on if CESM2(CAM6) represents that process, or how the results could differ it did.

- CESM2(CAM6) includes a representation of the entrainment process which could lead to decrease in LWP for an increase in aerosols through the unified turbulent scheme Cloud Layers Unified By Binormals (CLUBB). To clarify this, we added the following text in section 2.1:

  "*CESM2(CAM6) includes the unified cloudy turbulent scheme Cloud Layers Unified By Binormals (CLUBB) (Golaz et al., 2002). In CLUBB, cloud entrainment processes which could lead to deceased LWP are controlled by prognostic vertical turbulent fluxes and a tunable air parcel entrainment rate. However, both LWP and cloud cover are relatively insensitive to variation in the CLUBB parameter representing the entrainment rate (Guo et al., 2015). In their modelling study (not using CLUBB), Karset et al. (2020) further found that other factors, such as the sensitivity of the autoconversion rate to cloud droplet number concentration, play an even larger role in controlling the LWP than parameterized entrainment processes.*"

**2.3 Minor comments**

- P1 L2: Untangling rather than untangle makes more sense here.

  - "*Untangle*" has been changed to "*untangling*".

- P3 L75: Is x50 emissions a plausible scenario for an effusive eruption?

  - A scaling factor of x50 is indeed implausible for Icelandic effusive eruptions. We have replaced the sentence

    "*In addition to the ×1 scaling factor, corresponding to a Holuhraun-sized eruption, we perform simulations using scaling factors of ×5, ×25, and ×50, covering a plausible size range of Icelandic effusive eruptions.*"

    with

    "*In addition to the ×1 scaling factor, corresponding to a Holuhraun-sized eruption, we perform simulations using scaling factors of ×5 and ×25, covering the plausible range of Icelandic effusive eruptions, and ×50, extending into the size range of the largest known flood basalts on Earth (Kasbohm and Schoene, 2018).*"

- Figure 2 caption: I'm not clear what the ensemble mean refers to here if there is 1x simulation per start month and scaling as described in the methods.

  - Hoping to clarify our methodology, we replaced the sentence

    "*This results in an ensemble of ten sets of anomalies, one for each model year ...*"

    with

> "*This results in an ensemble of ten sets of anomalies for each combination of scaling factor and start month ...*"
>
> in the first line after Equation 2 (ca. line 86).

- P11 L188: I think 'we model' should be replaced with the 'model shows' or similar throughout as this is a result of the experiment rather than modelling a climate response explicitly.

    – We have changed "*we model*" to "*the model shows*" throughout the text.

- Figure 6: By eye it looks that there is less difference in the cloud properties and temperature response of eruptions starting due to starting in different months, and more that the impacts follow a seasonal cycle regardless of the start month of the eruption.

    – This is true and holds for the six months long eruptions we simulated. However, as illustrated in Figure 6f and discussed in Section 3.6, the model shows how temperature anomalies are carried across seasons. This means, for example, that if an eruption starting in June would continue throughout the winter, we might not get the positive temperature anomalies modelled for an eruption starting in December. For the six month long eruptions considered here, the starting month seems to be a considerable modulator to the temperature response. For eruptions lasting multiple years, the starting month is probably less important, and we would expect the impacts to rather follow the seasonal cycle.

- P15 L258: Wang et al. 2024 Hidden Large Aerosol-Driven Cloud Cover Effect Over High-Latitude Ocean discusses an Nd threshold for cloud fraction changes – might be of relevance here?

    – We thank the reviewer for bringing our attention to Wang et al.'s study and added a reference to it in our manuscript.

---

## Author Response (AR2)

**Authors' response**

We thank the anonymous referee and the editor, Dr. Khosrawi, for their helpful feedback, which have resulted in a much improved manuscript. We further thank them for their time. Below we include our reply (written in an upright font) to the comment of the referee (written in italics), and a full overview of changes made to the manuscript during the revision process.

Best regards,
the authors

**Anonymous Referee 1**

*The authors responded acceptably to most of my concerns. The exception is my major concern #2, which the authors did not respond to adequately. Indeed, they even have processed the observational data, so the least that could be done is including the observations in the analysis of the September eruption for the best-guess emission scenario, and discuss the conclusions from this.*

- We have addressed the referee's comment by adding a section titled "4.3 Observational evidence" to the Discussions section of the manuscript, along with a figure in the appendix (Figure A12). Below we include the text of the new Section 4.3 along with the new figure.

- **4.3 Observational evidence**

  The fall of 2014 was warm over the Greenland Sea (e.g., November in Ittoqqortoormiit, Jan Mayen, and Grímsey, Fig. A12b). Through observational and modelling evidence, Zoëga et al. (2023) argue that the 2014-15 Holuhraun eruption contributed to this warming signal through increased cloud LW trapping under limited sunlight. Although the simulations performed for this study here are not designed to exactly reproduce the 2014-15 Holuhraun eruption (for example with respect to the meteorology at the time) we do, nevertheless, see similarities in the climate response. When comparing anomalies from our `x1sep` simulations (which very closely resemble the 2014-15 Holuhraun eruption in terms of emissions and timing) averaged over the Greenland Sea (approximated by the area between 65°N to 80°N, and 25°W to 5°E) to anomalies from the ERA5 reanalysis for the same area, we find a warming signal in the fall months of September to November in both cases (Fig. A12a). This, along with the results of Zoëga et al. (2023), lends support to the credibility of the high-latitude winter warming mechanism discussed here.

[Figure]

**Figure A12:** *(a) Monthly mean surface air temperature anomalies averaged over the Greenland Sea as defined by the area between 66°N to 80°N, and 25°W to 5°E. The blue line shows the $x1sep$ CESM2(CAM6) ensemble mean anomalies. The black line shows anomalies in September 2014 to February 2015 relative to the 1984 to 2013 (30 years) climatology from the ERA5 reanalysis. Shades indicate 95 % confidence interval and 95 % prediction interval for the CESM2(CAM6) simulations and ERA5 respectively. (b) Observed monthly mean surface air temperature anomalies from stations in and around the Greenland Sea: Svalbard lufthavn/airport (78.25°N, 15.50°E; orange), Jan Mayen (70.94°N, 8.67°W; cyan), Ittoqqortoormiit (70.48°N, 21.95°W; purple), Danmarkshavn (76.77°N, 18.68°W; olive), and Grímsey (66.54°N, 18.02°W; green). Anomalies are for September 2014 to February 2015 relative to the 1984 to 2013 climatology. Prediction intervals are not plotted to preserve clarity. For all timeseries in both (a) and (b), filled dots indicate significant anomalies, open dots insignificant.*

**Overview of changes**

We made a few additional changes to the manuscript to accommodate for the new Section 4.3 and Figure A12, and fixed a few technical things pointed out by Dr. Khosrawi. A full list of changes is included below. Line numbers refer to the newly revised manuscript.

- Lines 95 to 98: We added a new section titled "2.3 Observations and reanalysis" where we briefly describe the data we used to create Figure A12, and add the respective references.

- Line 107: We added a missing comma after "e.g."

- Lines 112 to 14: We added a brief description of the processing of the reanalysis and observational data.

- Line 199: We added a line break to avoid separation of number and unit.

- Line 313: We changed "Section" to "Sect."

- Lines 363 to 346: We added a section titled "4.3 Observational evidence".

- Lines 389 to 391: We added a few lines to the code and data availability statement to acknowledge the use of the reanalysis and observational data. This change does for some reason not appear in the track-changed version of our manuscript.

- Line 408: We added a figure in the appendix (Figure A12).

---

## Author Response (AR3)

**Authors' response**

We thank the editor, Dr. Khosrawi, for handling our manuscript and the anonymous referees for their helpful feedback. We further thank them for their time. After the manuscript was accepted for publication, we updated the code and data availability statement (starting in line 388) by including a reference to a data archive containing the model output underlying the results of the study. We further added the corresponding entry to the list of references. The updated code and data availability statement reads as follows:

> *The CESM2(CAM6) output underlying the results and figures presented in this paper, along with a Jupyter Notebook containing plotting scripts for the figures, are available at the NIRD Research Data Archive (Zoëga, 2025). The ERA5 reanalysis is available at the Copernicus Climate Change Service (C3S) Climate Data Store (CDS) (Hersbach et al., 2024). Observational timeseries are available at NCCS (2023) for Svalbard lufthavn/airport and Jan Mayen, DMI (2023) for Danmarkshavn and Ittoqqortoormiit, and Icelandic Met Office (2024) for Grímsey.*

Best regards,
the authors